# Engineering a passivating electric double layer for high performance lithium metal batteries

Weili Zhang[1], Yang Lu[1], Lei Wan[1], Pan Zhou[1], Yingchun Xia[1], Shuaishuai Yan[1], Xiaoxia Chen[1], Hangyu Zhou [1], Hao Dong[1] & Kai Liu [1]✉

In electrochemical devices, such as batteries, traditional electric double layer (EDL) theory holds that cations in the cathode/electrolyte interface will be repelled during charging, leaving a large amount of free solvents. This promotes the continuous anodic decomposition of the electrolyte, leading to a limited operation voltage and cycle life of the devices. In this work, we design a new EDL structure with adaptive and passivating properties. It is enabled by adding functional anionic additives in the electrolyte, which can selectively bind with cations and free solvents, forming unique cation-rich and branch-chain like supramolecular polymer structures with high electrochemical stability in the EDL inner layer. Due to this design, the anodic decomposition of ether-based electrolytes is significantly suppressed in the high voltage cathodes and the battery shows outstanding performances such as superfast charging/discharging and ultra-low temperature applications, which is extremely hard in conventional electrolyte design principle. This unconventional EDL structure breaks the inherent perception of the classical EDL rearrangement mechanism and greatly improve electrochemical performances of the device.

[1] Department of Chemical Engineering, Tsinghua University, Beijing, China. ✉email: liukai2019@tsinghua.edu.cn

The electric double layer (EDL) is the region where all electrochemical reactions occur, and its properties determine the process of the electrochemical reaction at the electrode/electrolyte interface[1, 2]. However, for a long time, a great deal of research on electrolyte design in electrochemical devices has focused on the regulation of cation or anion solvation structure in bulk electrolyte, because the classical theory that bulk electrolyte structure determines EDL properties. These strategies are generally difficult to achieve a perfect balance among all the required performance of the energy storage system[3–5].

Specifically, for lithium metal batteries, which is well-recognized as the next-generation energy storage devices, the development of suitable electrolytes for lithium metal batteries coupled with high-voltage (>4.0 V vs. Li/Li[+]) cathode materials face a dilemma[6–9]. Among the studies reported so far, carbonates and ethers are two most popularly employed solvents and show the best competitive comprehensive properties. On the one hand, carbonate electrolyte with high oxidation stability is highly corrosive to lithium metal, and porous SEI derived from carbonate solvent will lead to the continuous decomposition of solvent and the rapid failure of battery during lithium metal deposition and stripping[10, 11]. On the other hand, ethers exhibit several significant advantages compared with carbonates, such as excellent compatibility with lithium metal[12], superior Li[+] transport dynamics due to the extremely low viscosity[13, 14], and ultralow freezing point guarantee battery cycling at low temperatures[15, 16]. Though with the above advantages, unfortunately, the poor oxidation stability (<4.0 V) greatly limits its application in high-voltage batteries. Ethers with a typical salts concentration of 1 M cannot be used with nickel-rich (such as $LiNi_{0.8}Mn_{0.1}Co_{0.1}O_2$, NMC811) or other high-voltage cathodes, therefore greatly hindered their application in high-energy density lithium metal batteries. Several methods have been employed to solve this problem. Recently, "solvent-in-salt" has received wide attention, in fact by greatly increasing the concentration of the bulk electrolyte, forcibly reducing the content of the free solvent in the EDL to achieve the purpose of improving the electrochemical stability of overall electrolyte. However, this method inevitably leads to a series of problems such as high viscosity, high cost and poor dynamic behavior[17–19]. Although the localized high-concentration electrolyte is developed to solve this problem by diluting the high-concentration electrolyte with inert solvents, the diluent used is actually insoluble to the lithium salt, and there may be a risk of lithium salt precipitation at ultralow temperatures, thus deteriorating the low-temperature performance of the

battery[20–22]. It is necessary to mention that high-voltage additives such as lithium difluoro(oxalato)borate (LiDFOB) can decompose at the cathode–electrolyte interface before electrolyte decomposition and participate in the formation of dense interfacial film (CEI layer) to suppress the decomposition of electrolyte[23–25]. Unfortunately, The CEI passivation layer does not essentially alter the electrochemical stability of EDL and therefore cannot completely prevent the degradation of unstable EDL at the interface, resulting in the continuous thickening of CEI during the cycle[20, 26].

Therefore, if a highly stable EDL (Li[+]-rich state with free solvents well-coordinated) formed in situ at the electrode/electrolyte interface, it can decouple the high-voltage cathode and the bulk electrolyte without losing the intrinsic properties of the electrolyte. Unfortunately, it is well known that keeping the EDL at cathode/electrolyte interface in a Li[+] rich state during charging is irreconcilable with the traditional EDL rearrangement theory. Therefore, in order to solve the above contradiction, we propose a new EDL rearrangement mechanism to stabilize ether solvent molecules at the cathode/electrolyte interface by selecting suitable voltage stimulation responsive additives to bind Li[+] and free solvents firmly inside the EDL during charging, forming a special EDL that could passivate the continuous anodic decomposition of the bulk solvents (Fig. 1a). Thus, this innovative EDL design qualitatively changes the oxidation stability of the overall electrolyte, enabling the ether-based electrolyte to be successfully used in high-voltage lithium metal batteries with greatly suppressed anodic decomposition and fully release its advantages such as superfast charging capacity and ultra-low-temperature performance.

## Results

**Design mechanism of EDL and screening for additives.** On the one hand, this voltage-stimulated response additive is expected to migrate rapidly to the electrode surface driven by the electric field, which corresponds to the ionic charge density that can be quantitatively estimated by calculating the ion size (Supplementary Table 1). More importantly, this additive requires a strong ability to bind Li[+]. Here, nuclear magnetic resonance (NMR) spectroscopy is implemented to judge the ability of the additives to coordinate Li[+], because if the additives have a strong ability to attract Li[+], Li[+] will tend to accumulate in the solution, resulting in a stronger de-shielding effect[27] (Supplementary Fig. 1). From the above analysis, an anion with a small size and a strong ability

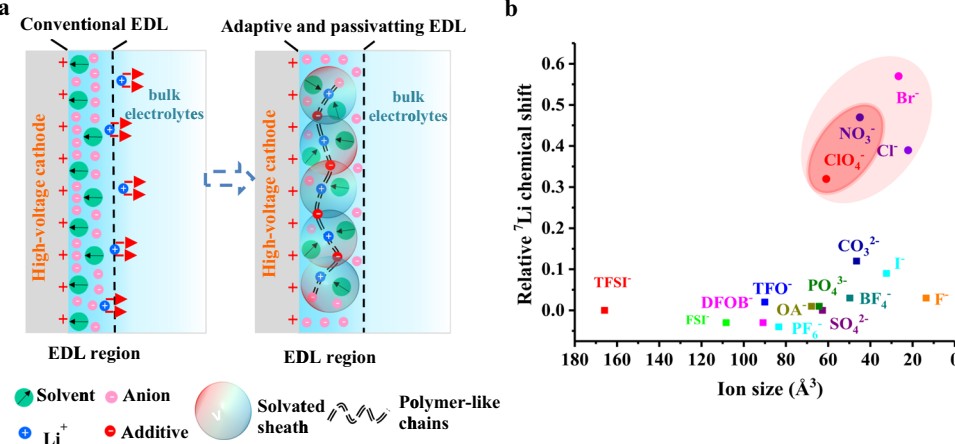

**Fig. 1 EDL design strategy and selection of additives. a** Proposed mechanism for constructing a dynamic high-voltage resistance EDL. **b** The selection of stimulus-response additives depends on ion volume and relative [7]Li chemical shift.

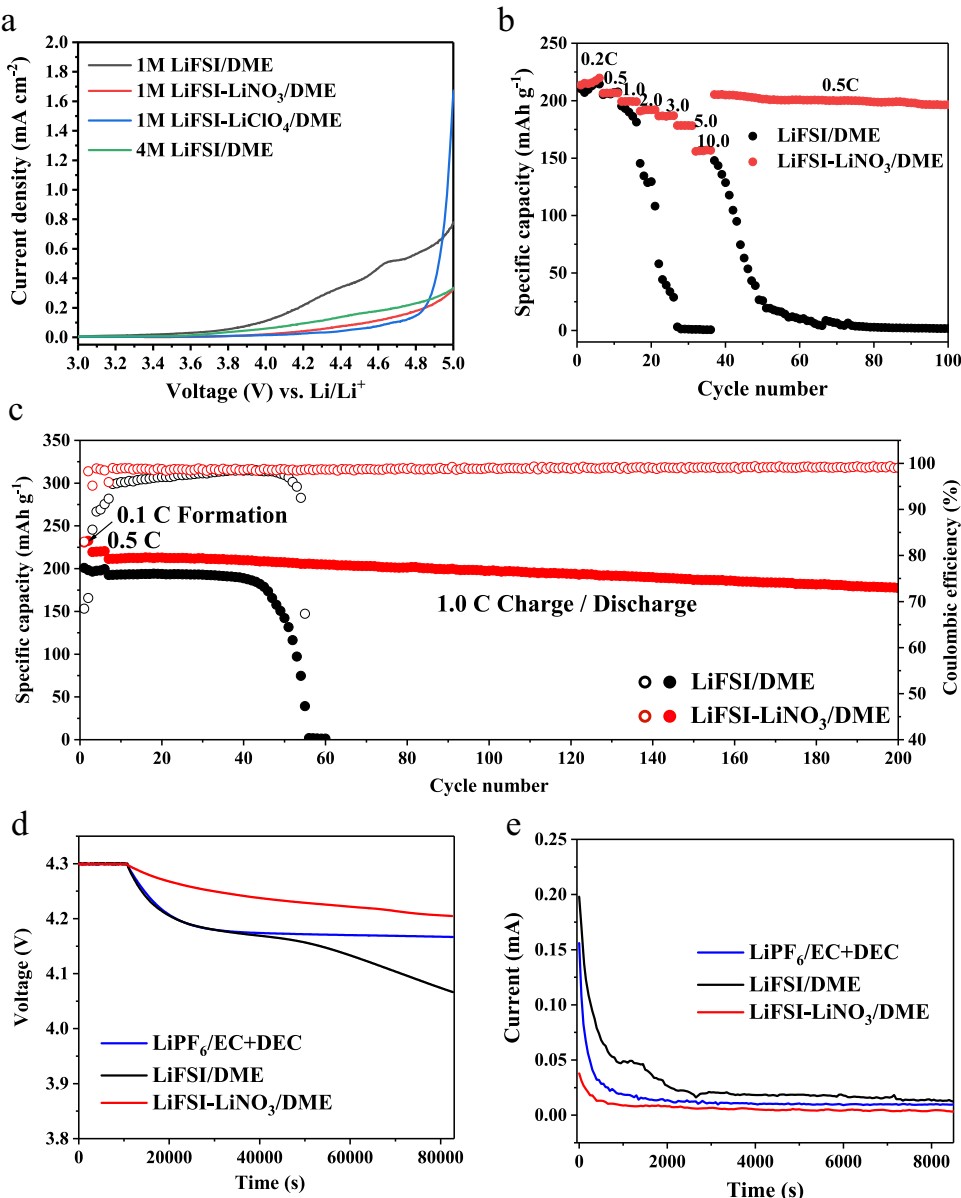

**Fig. 2 Comparison of electrochemical properties of LiFSI/DME and LiFSI–LiNO₃/DME. a** Oxidative stability measured via LSV for Li||Al cells. **b** Rate capability of Li||NMC811 cells under different charging/discharging rates. **c** Long-term cycle performance of Li||NMC811 cells using different electrolytes at 1.0 C rate. **d** Self-discharge tests after a potentiostatic hold at 4.3 V vs Li/Li⁺. **e** Typical current relaxation curves collected from Li||NMC811 half-cells during a potentiostatic hold at 4.3 V vs Li/Li⁺.

to coordinate Li⁺ would be a perfect choice. Selecting from a range of commonly used anions, we identified four potential candidates, i. e. $NO_3^-$, $ClO_4^-$, $Br^-$, $Cl^-$, as shown in Fig. 1b. But for well-known reasons, $Br^-$ and $Cl^-$ are excluded because they are extremely active and easily oxidized at high voltage to produce products unfavorable to the battery system.

**Electrochemical performance and characterization of the electrolyte**. First, we verify whether the two candidates can improve the anodic stability of ether-based electrolyte, as presented in Fig. 2a, the degradation potential (take 0.05 mA cm⁻² as the criterion) of 1 M Lithium bis(fluorosulfonyl)imide/1,2-dimethoxyethane (LiFSI/DME) is earlier than 4.0 V (vs Li/Li⁺), while the presence of 0.3 M LiNO₃ broadens the electrochemical stability window to 4.3 V, and the decomposition potential is further widened to 4.5 V with the addition of 0.3 M LiClO₄, which is even more significant than the effect of the means of high-

concentration electrolytes (4.2 V vs Li/Li⁺ for 4.0 M LiFSI/DME). Next, we take 0.3 M LiNO₃ additive as an example to investigate the compatibility of LiFSI–LiNO₃/DME with high-voltage cathode materials (NMC811) and verify the proposed voltage responsive mechanism. Both rate and cycling tests ranging from low rate (0.1 C) to high rate (10.0 C) were conducted for NMC811 electrodes in LiFSI/DME and LiFSI–LiNO₃/DME electrolytes. As shown in Fig. 2b, as expected, LiFSI/DME suffers rapid capacity fading with increasing current. On the contrary, the LiFSI–LiNO₃/DME exhibits a remarkable rate performance, achieving a discharge capacity of 152 mAh g⁻¹ even at a high rate of 10.0 C. Moreover, the cell can still maintain a stable cycle after returning from the rate of 10.0 C to the rate of 0.5 C, which proves the sustainable adaptability of LiFSI–LiNO₃/DME to the highly active NMC811 cathode material. It is worth noting that the coulombic efficiency of LiFSI/DME in formation cycle is only ~68.44%, while that of LiFSI–LiNO₃/DME in the formation cycle

is significantly increased to 85.98% (Supplementary Fig. 2). In addition, the long cycle tests of LiFSI–LiNO$_3$/DME at different rates are extremely stable (Fig. 2c, and Supplementary Figs. 3–8), while LiFSI/DME is subject to continuous capacity at all rates, demonstrating the important role of LiNO$_3$ in improving oxidation-resistance of DME-based electrolytes. Some other additives have also proved to be ineffective in improving cycle stability (Supplementary Fig. 9). The reactivity of the different electrolytes with NMC811 cathodes was further investigated by recording the open-circuit voltage (OCV) of a fully charged cells as a function of storage time. As shown in Fig. 2d, the OCV of a fully charged cells with LiFSI/DME drastically decreased after circuit disconnection, whereas a cell charged in LiFSI–LiNO$_3$/DME exhibited relatively little OCV change, even better than conventional commercial electrolytes. Figure 2e shows the leakage current as a function of time generated by a potentiostatic hold (4.3 V) for cells with different electrolytes. The initial decrease in the curve is the relaxation of the concentration gradient in the cell, and the final static leakage current value is usually used as an indicator to quantify the rate of the parasitic reaction. Obviously, LiFSI/DME shows a much higher static leakage current than the other two electrolytes, indicating a faster parasitic reaction between electrode and electrolyte. The presence of NO$_3^-$ significantly improves the stability of the electrode/electrolyte interface, thereby effectively diminishing reaction rate and inhibiting the self-discharge of a fully charged DME-based cells. The extensive intergranular cracking in the NMC811 cathode cycled with LiFSI/DME (Supplementary Fig. 10a) means more side reactions as well as faster electrolyte consumption that will eventually result in a cliff-like decay of capacity. However, the electrode particles cycled in LiFSI–LiNO$_3$/DME system preserved mechanical integrity without visible cracks (Supplementary Fig. 10b), further supporting superior compatibility and stability of this electrode/electrolyte interphase.

Moreover, compared to commercial carbonate electrolytes, DME-based electrolytes exhibit significantly higher conductivity at all tested temperatures, as well as higher Li$^+$ transference number. As shown in Fig. 3a, the conductivity of the DME-based electrolyte is observed to retain a remarkable value (3.68 mS cm$^{-1}$ for LiFSI/DME, 2.87 mS cm$^{-1}$ for LiFSI–LiNO$_3$/DME) even at −60 °C are attributed to the low melting point (~−69 °C), compared with the values less than 0.1 mS cm$^{-1}$ in LiPF$_6$/ EC + DEC at temperatures below −20 °C. The Li$^+$ transference number of DME-based electrolyte reported in Fig. 3b all exceed 0.4 (0.45 for LiFSI/DME, 0.43 for LiFSI–LiNO$_3$/DME), while that of commercial carbonate-based electrolytes is only 0.31. Therefore, after solving the problem of poor oxidation stability of DME-based electrolyte, the high-voltage lithium metal battery with DME-based electrolyte and NMC811 cathode can be expected to achieve faster charge/discharge rates and lower operating temperatures while obtaining high-energy density at the same time. As shown in Fig. 3c, the LiFSI–LiNO$_3$/DME system maintained a high discharge capacity of 150 mAh g$^{-1}$ (69.7% of the capacity at 0.2 C rate) at 10.0 C rate, whereas the capacity of the 1 M LiPF$_6$/EC + DEC system was limited to only 30 mAh g$^{-1}$ (15% of the capacity at 0.2 C rate). The three-electrode experiments are implemented to decouple the anode and cathode contributions to the overpotential at different rates. As shown in Supplementary Fig. 11, the overpotential of lithium metal anode in ether-based electrolytes is much smaller than that of carbonate-based electrolytes at low rates (0.2 C, 0.5 C) due to the superior compatibility between lithium metals and ether solvents, but there is no significant difference in the overpotential of the two electrolytes on the cathode side. However, with the increase of the charge/discharge rate, the struggling dynamic behavior of the carbonate-based electrolyte on the cathode side

dominates the performance loss. Therefore, the excellent fast charging performance of Li||NMC811 cells with LiFSI–LiNO$_3$/ DME is mainly attributed to the fast kinetic response of our electrolyte on the cathode side. Moreover, Fig. 3d and e demonstrated the sustainable superfast charging/discharging capability of the LiFSI–LiNO$_3$/DME system, which maintains 90% capacity after 700 cycles at 5.0 C charge/discharge rate, while the cell with commercial carbonate electrolyte systems suffered from severe capacity decay accompanied by unstable coulombic efficiency. It is well known that the LiNO$_3$ additive could make the electrochemical behavior of Li metal anode even better (Supplementary Fig. 12). Thus, the high-voltage Li||NMC811 full cells (thin Li anode: 40 μm, high-loading cathode NMC811:20.3 mg cm$^{-2}$, the N/P ratios of the Li||NMC811 cell was 2.31) were tested to further explore utility function of the LiFSI–LiNO$_3$/ DME system under highly challenging conditions. As shown in Fig. 3f, Li||NMC811 full cells using LiFSI–LiNO$_3$/DME system achieved absolute superior cycling performance even at a high rate of 2.0 C with 82% capacity retention after 200 cycles (Fig. 3g). To verify the universality of the proposed mechanism discussed above, we prove that the LiFSI–LiNO$_3$/DME system can successfully pair with another high-voltage cathode LiCoO$_2$ (Supplementary Fig. 13a, b). Further, we replace the 0.3 M LiNO$_3$ with 0.3 M LiClO$_4$, and found that the LiFSI-LiClO$_4$/DME system can also pair NMC811 cathodes and demonstrate superior cycling performance at a high cutoff voltage of 4.5 V (Supplementary Fig. 13c, d). In addition, we also confirmed that (LiFSI–LiNO$_3$/ Tetrahydrofuran) LiFSI–LiNO$_3$/THF system can be successfully applied to Li||NMC811 cells by replacing DME with THF (Supplementary Fig. 13e, f). Moreover, ether solvents generally exhibit extremely low freezing point and low viscosity, making it one of the most ideal candidates for low-temperature battery[14, 15]. As shown in Supplementary Fig. 14a–c, in contrast to the notorious poor low-temperature performance of commercial electrolytes, LiFSI–LiNO$_3$/DME can be stably cycled 200 times at −20 °C. Extensive previous studies suggest that low-temperature performance depends largely on the charge transfer step, similarly demonstrated by our three-electrode experiments. Specifically, as shown in Supplementary Fig. 15, the LiFSI/DME electrolyte shows significantly lowered charge transfer resistance than LiPF$_6$/ EC + DEC electrolyte, and the LiFSI–LiNO$_3$/DME further lowered this value. The kinetics of different interfacial processes were further studied by temperature-dependent electrochemical impedance spectroscopy (EIS) (Supplementary Fig. 16). The activation energies of each interfacial process on the cathode side were obtained by fitting EIS spectra according to the classical Arrhenius law. LiFSI/DME shows a slightly reduced activation energy (Ea,ct = 50.21 kJ mol$^{-1}$) compared to LiPF$_6$/EC + DEC (Ea,ct = 53.97 kJ mol$^{-1}$). Interestingly, the addition of LiNO$_3$ further weakened the activation energy (Ea,ct = 44.23 kJ mol$^{-1}$). Therefore, we believe that the better low-temperature performance of DME-based electrolyte than carbonate-based electrolyte is the result of the integrated contribution of bulk transport as well as interfacial transport. Moreover, the altered EDL due to NO$_3^-$ even slightly accelerates the Li$^+$ charge transfer behavior of DME-based electrolytes. More surprisingly, the Li|| NMC811 cells using LiFSI–LiNO$_3$/THF can even operate at an extreme temperature of −91 °C, delivering a capacity of 86 mAh g$^{-1}$ (Fig. 3i), attributing to the extremely low freezing point of THF (−108.5 °C) and higher conductivity at ultra-low temperature (Supplementary Fig. 17). As a result, in ultralow temperature (−40 °C) long-term cycling, Li||NMC811 cells with LiFSI–LiNO$_3$/THF shows excellent stability, providing an initial capacity of 160 mAh g$^{-1}$ and 82% capacity retention after 150 cycles (Supplementary Fig. 14d). Further, the LiFSI–LiNO$_3$/THF-

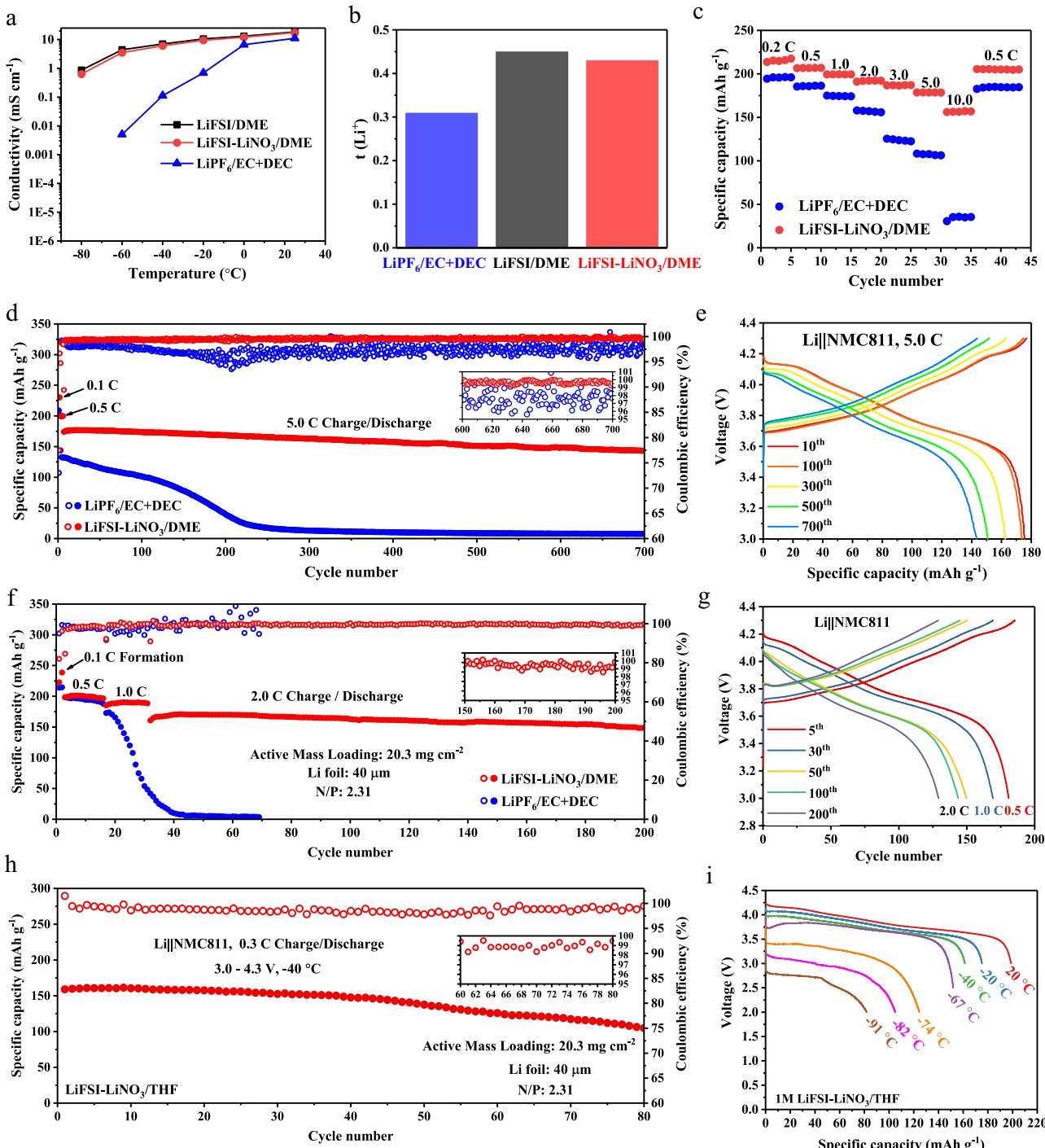

**Fig. 3 Comparison of transport and electrochemical properties between the ether-based electrolyte and carbonate-based electrolyte. a** Conductivity versus temperature. **b** Li[+] transference number (t [Li+]) computed from DC polarization measurements at 10 mV using the Bruce–Vincent method. **c** Rate capability of Li||NMC811 cells under different charging/discharging rates. **d** Long-term superfast charging/discharging performance of Li||NMC811 cells using different electrolytes at 5.0 C rate. **e** the corresponding voltage profiles at different cycles of Li||NMC811 cells using LiFSI–LiNO$_3$/DME at 5.0 C rate. **f** Long-term cycling performances of high-voltage Li||NMC811 full batteries with 40 μm Li anode. The N/P ratios of the Li||NMC811 cell were 2.31. The first two formation cycles were carried out at a 0.1 C rate, followed by 15 cycles at 0.5 C rate, sequential 15 cycles at 1.0 C rate and the long-term cycling was at 2.0 C rate. **g** The corresponding voltage profiles of high-voltage Li||NMC811 full batteries using LiFSI–LiNO$_3$/DME. **h** Cycling performance of full cells in LiFSI–LiNO$_3$/THF electrolyte at −40 °C and 0.3 C rate. **i** Discharge profiles of Li||NMC811 cells using LiFSI–LiNO$_3$/THF electrolyte at different temperatures.

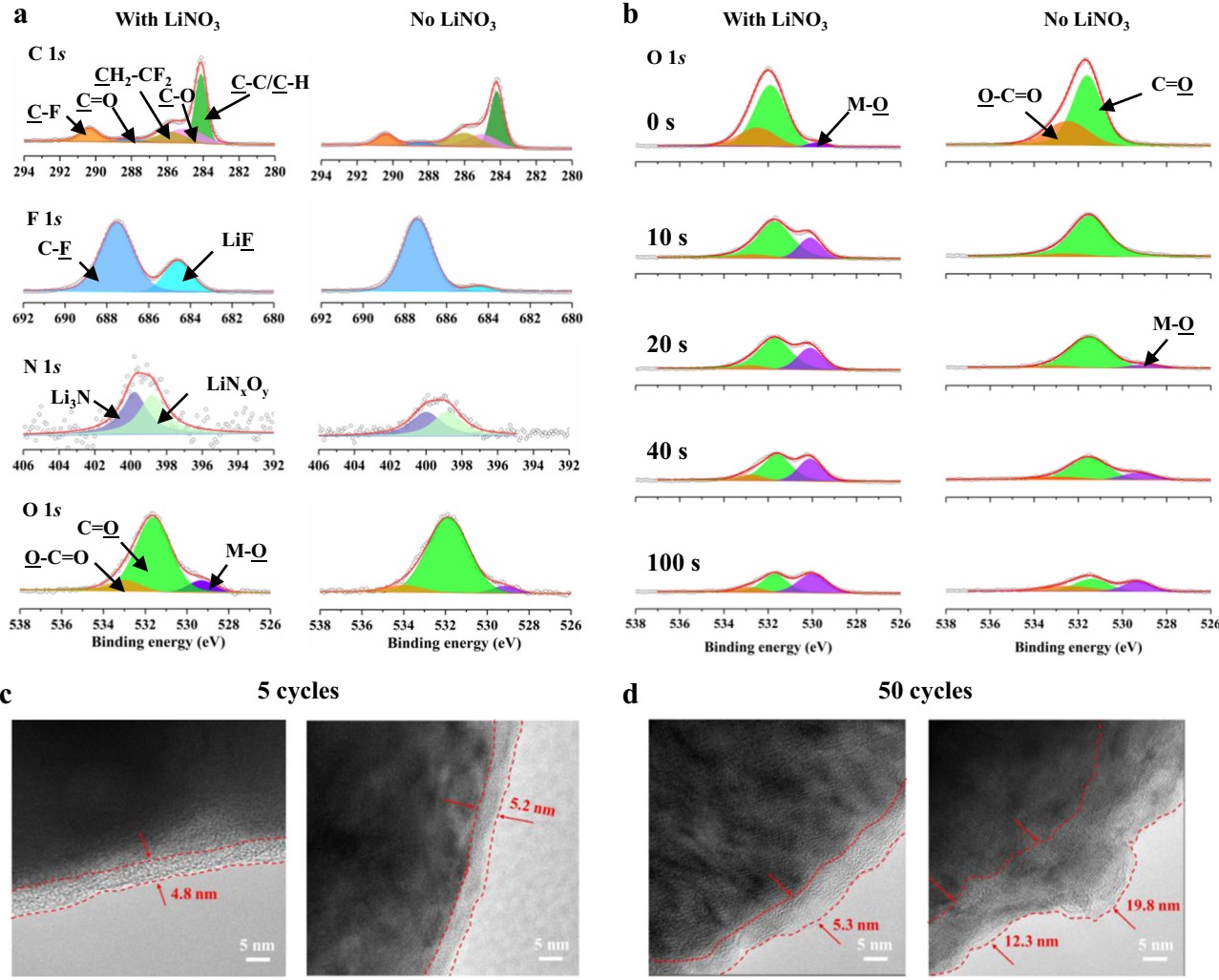

**Fig. 4 Components on the surface of cathodes after cycling. a** X-ray photoelectron spectroscopy (XPS) profiles of C 1s, F 1s, N 1s, and O 1s of CEI formed on NMC811 surface after 5 cycles and **b** XPS depth profiles of O 1s of CEI formed on NMC811 surface after 50 cycles at 0.5 C in Li||NMC811 coin cells with LiFSI/DME and LiFSI–LiNO₃/DME electrolyte. HRTEM analyses of NMC811 cathodes obtained from Li/NMC811 cells using LiFSI/DME (right) and LiFSI–LiNO₃/DME (right) after (**c**) 5 and (**d**) 50 cycles.

based full cells were cycled at −40° to provide the holistic effect of both the lithium metal anode and the NMC811 cathode at ultralow temperature. As displayed in Fig. 3h, stable battery cycle in full cells can be achieved using LiFSI–LiNO₃/THF electrolytes, retaining 68% of the initial areal capacity after 80 cycles, demonstrating the compatibility of the electrolyte system with the cathode and anode and the reversibility of the electrochemical process at ultralow temperature. As far as we know, there is no report so far that a lithium metal battery system can combine high cutoff voltage, fast charging capability, and ultra-low temperature performance[14, 28–30] (Supplementary Fig. 18 and Supplementary Table 3).

**Interface between cathode and electrolyte.** To understand the role of NO₃⁻ in enhancing the oxidative stability of ether-based electrolytes, X-ray photoelectron spectroscopy (XPS) analysis was carried out to explore the difference of solid electrolyte interphase on the cathode surface (CEI) after five cycles in LiFSI/DME and LiFSI–LiNO₃/DME system. As shown in Fig. 4a, in the LiFSI/DME electrolyte system, organic and inorganic composite components derived from the decomposition of ether-based electrolytes, such as C–C/C–H, C–O, C = O, CH₂–CF₂, LiF, Li₃N, and LiNₓOᵧ, are observed on the top surface of the cycled NMC

cathodes. Apparently, this kind of CEI components could not effectively impede the continued decomposition of solvents in LiFSI/DME systems. Unexpectedly, we also found C-C/C–H/C–O species in the CEI of the LiFSI–LiNO₃/DME system, indicating that DME molecules are also involved in the formation of the CEI, but its content is significantly lower than that of the LiFSI/DME system. Higher LiF content was detected on the top surface of the cycled NMC811 cathodes in LiFSI–LiNO₃/DME system as shown in F 1s, proving that LiNO₃ promoted the decomposition of FSI⁻ anions to form more LiF. The presence of more LiF, which is known for its excellent electronic insulation, may be able to prevent electrons from tunneling through the CEI, thereby preventing the continuous decomposition of the electrolyte. In general, no significant component differences were found on the top surface of the CEI formed in the two electrolytes, except for slight variations in composition concentration and higher LiF content. We further measured the thickness evolution of the CEI after 5 cycles and 50 cycles in the two electrolytes by depth sputtering profiles. In Fig. 4b, the signal of the metal oxide bond (M–O) (from NMC) is observed at the first sputtering, indicating that very thin CEI layer is formed after five cycles in both electrolytes. But after 50 cycles, the signal of the metal oxide bond (M–O) (from NMC) still appeared at the first sputtering in the

LiFSI–LiNO$_3$/DME system. However, for the neat LiFSI/DME system, the M–O bond signal is not detected until the sputtering depth reaches 10 nm. The TEM image is consistent with the XPS test, as shown in Fig. 4c, the thickness of the CEI layer on the surface of the NMC811 after 5 cycles in both electrolytes is about 5 nm. However, after 50 cycles, the thickness of the CEI layer formed in LiFSI/DME increased dramatically to 12–20 nm (Fig. 4d), indicating that the initial formed CEI could not effectively protect the electrode/electrolyte interface. In contrast, the appearance of LiNO$_3$ successfully inhibits the continuous degradation of electrolytes, preserving uniform and thin CEI layers during the cycle, which should be the main reason for the improved electrochemical performances of LiFSI–LiNO$_3$/DME. Is these slight variations in the composition of CEI mainly responsible for the suppressing of the side reaction between ether molecules and NMC cathode? To answer this question, we cycled the NMC cathode in the LiFSI–LiNO$_3$/DME system two times to form CEI, and then replaced the electrolyte with LiFSI/DME, but the reassembled cells cannot operate efficiently, and the capacity decays rapidly after 50 cycles, which strongly demonstrates that the passivation CEI layer alone cannot prevent the decomposition of ether molecules (Supplementary Fig. 19). In addition, we also excluded the influence of electrolyte concentration, as shown in Supplementary Fig. 20, the cell with 1.3 M LiFSI/DME also failed to last more than 50 cycles. But, if LiFSI is replaced with LiBF$_4$, the cell can work stably and efficiently, as shown in Supplementary Fig. 21. Therefore, the above results confirm the key role of NO$_3^-$ in protecting solvents from decomposition and exclude the main responsibility of CEI.

**Dissection of EDL by molecular dynamics simulation.** Does the structural change of the interface EDL improve the voltage resistance of the DME-based electrolyte? We first dissect the EDL structure at the electrode/electrolyte interface by MD simulation and DFT calculation[31, 32]. Typically, with the increase of the repulsive force from the electric field, positive ions will be gradually excluded from the highly polarized electrode surface, as observed in Fig. 5a, the concentration of the Li$^+$ layer closest to the electrode interface in the LiFSI/DME system did drop sharply as expected and farther away from the electrode, thus leaving more free DME molecules with a potential risk of degradation at high voltages. However, although Li$^+$ expulsion still occurs in the LiFSI–LiNO$_3$/DME system, the degree of this reduction is greatly alleviated relative to the LiNO$_3$-free system, and the position of Li$^+$ layer close to the electrode interface almost does not change, owing to the strong electrostatic interaction of NO$_3^-$ with Li$^+$ (Fig. 5b). Figure 5c shows a more distinct comparison of the content of Li$^+$ in the interfacial region as the potential increases. In addition, we divided the DME adsorbed in the double layer into bound and free states based on whether it is within 0.3 nm distance around Li$^+$ ions. The number density profiles of bound DME in Supplementary Fig. 22a and b suggest that the variation trend of bound DME in the EDL with voltage is almost consistent with that of Li$^+$ ions, that is, the decline of bound DME in the LiFSI–LiNO$_3$/DME system is obviously more moderate than that in the LiFSI/DME system, because more Li$^+$ ions will bind more DME. Similarly, by counting the bound DME content in the EDL under each voltage, we can see the huge difference of the environment of DME in the EDL (Fig. 5d). A comparison of local structure of inner-Helmholtz interfacial regions at cathode surface in two electrolyte systems from molecular dynamics (MD) simulations is given in Fig. 5e and f (Supplementary Figs. 23 and 24). For the LiFSI/DME system, when the voltage is polarized from the PZC to 0.5 V, the anion is adsorbed and enriched while the lithium ion is greatly expelled from the EDL inner layer and

only a few lithium ions can be seen in the field of vision, resulting in a large number of DME in a free state. By contrast, even if the positive potential is applied, a large amount of lithium ions is still retained in the EDL inner layer of LiFSI–LiNO$_3$/DME system. As shown in Fig. 5f, most DME coordinated by Li$^+$ ions, and multiple polymer-like chains structures are formed at the electrode interface (as highlighted by the yellow region in Fig. 5f). Consequently, the DME molecules we collect at the interface in the LiFSI–LiNO$_3$/DME system are more involved in the Li$^+$ centered cluster structure instead of in the form of free state. Interestingly, for the collected clusters with the presence of anions and solvents (Li-DME-FSI, Li-DME$_2$-FSI, Li-DME-NO$_3$, Li-DME$_2$-NO$_3$) (Supplementary Fig. 25), the highly active site shifted from DME to FSI$^-$, indicating that anions in this cluster are preferentially decomposed to generate popular inorganic products such as LiF and Li$_3$N, which is consistent with the XPS results shown above. Overall, the above calculation results confirm that the introduction of NO$_3^-$ constructs a Li-rich-double layer interface at the positive potentials of the electrode in the early stage of charging, forming unique polymer-like chains structures with high electrochemical stability in the EDL inner layer. This will act as a shielding network between the highly active cathode and the diffusion layer of the electrolyte, preventing the bulk DME electrolyte from being continuously degraded at high voltages. For comparison, additional calculations were performed using BF$_4^-$ and SO$_4^-$ as additives. As shown in Supplementary Fig. 28, the presence of SO$_4^-$ and BF$_4^-$ did not significantly increase the proportion of bound DME at the interface, and the polymer-like chain structure appearing in the NO$_3^-$ system is not observed in the inner-Helmholtz interfacial regions of these two systems (Supplementary Figs. 29 and 30), thus further verifying the unique properties of NO$_3^-$ in changing the EDL structure.

**Dissection of EDL by in situ EC-SERS and in situ EC-AFM.** To explore the evolution of the EDL from the experimental point of view, we combined vibration spectroscopy and electrochemical methods to track the species and molecular specificity at the interface. First of all, electrochemical surface-enhanced Raman spectroscopy (EC-SERS) is a powerful technique to detect the vibration fluctuation of surface matter during the electrochemical reactions[33, 34]. In this experiment, we detect the structural dynamics of molecules and ions at the interface in situ by depositing a layer of gold nanoparticles on the aluminum mesh as a Raman enhanced substrate (Fig. 6a). As shown in Fig. 6b and c, the in situ EC-SERS spectra of the two electrolytes system at the interface with charging were obtained. Figure 6d shows the ions and solvent concentrations at the interface of two electrolyte systems as a function of voltage by calculating the area of each component in Fig. 6b and c. The Raman peaks at 700–760 cm$^{-1}$ arise from the vibration of the FSI$^-$ anions[35–37], and the FSI$^-$ ion concentration at the interface rises gradually with the increase of the voltage in the two electrolytes system, indicating the enrichment of the counterion under the effect of the electric field. Obviously, in the LiFSI–LiNO$_3$/DME system, a new Raman peak belonging to the NO$_3^-$ appears at 1040 cm$^{-1}$, which is also enriched at the interface with the increase of voltage, and can be well understood by the enhanced coulombic interaction between counterions and electrode[38]. Moreover, there are several Raman bands at 800–900 cm$^{-1}$ in DME solvent. The two bands at 821 cm$^{-1}$ and 849 cm$^{-1}$ should be attributed to the free DME solvent with various isomers, and the band at 879 cm$^{-1}$ is from Li$^+$-solvated DME solvent[39]. For the electrolyte without LiNO$_3$, the majority of DME molecules at the interface exist in the form of free state, and the proportion of Li$^+$-solvating DME decreases with positive polarization, thus

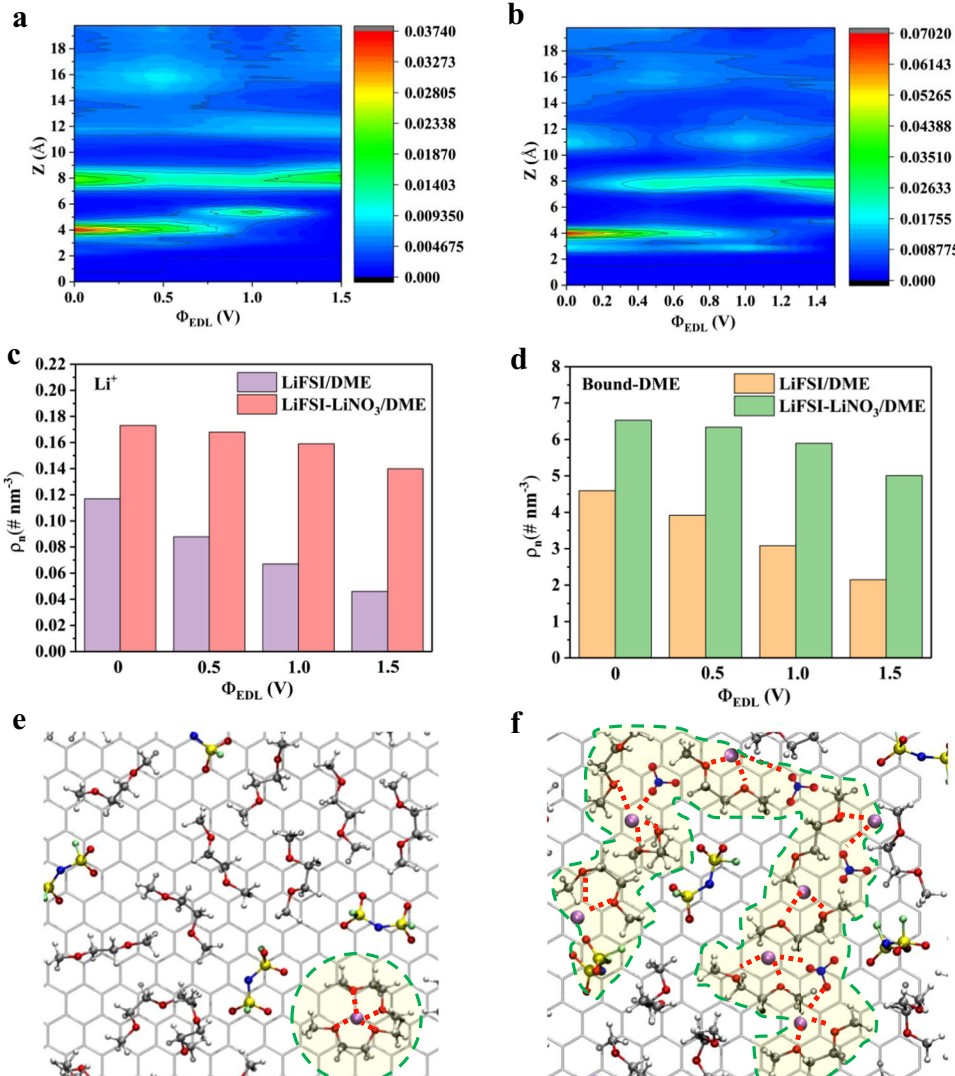

**Fig. 5 Molecular dynamics simulation of EDL structure.** Number densities of Li+ in (**a**) 1 M LiFSI/DME and (**b**) LiFSI–LiNO₃/DME as a function of distance from the graphite electrode (z) at various potentials (ΦEDL). The color scale indicates the number of densities of Li+. Comparison of number densities of the (**c**) Li+ and (**d**) bound DME in the interfacial region at different electrolytes system. Local structure of inner-Helmholtz interfacial regions at cathode surface in (**e**) 1 M LiFSI/DME and (**f**) LiFSI–LiNO₃/DME at 0.5 V.

accelerating the degradation of DME. However, for LiFSI–LiNO₃/DME system, the proportion of Li+-solvating DME solvents present the obviously increasing trend as positive polarization, which indicates that the introduction of NO₃⁻ makes the DME more involved in the solvation shell sheath, indicating that the concentration of Li+ at the interface is not greatly reduced by repulsive force of electric field at the beginning of charging, consistent with our proposed design mechanism of EDL.

Electrochemical atomic force microscopy (EC-AFM) can visualize the topography of electrode/solution interface from the molecular level to atomic resolution in a vacuum or immersed in liquid[40–43]. Here, we performed EC-AFM experiments on a sealed electrochemical cell with highly oriented pyrolytic graphite (HOPG) to provide atomic-scale details of the interfacial organization of the EDL structures of our designed electrolytes system using the bluedrive photothermal excitation system (Fig. 7a). As shown in Fig. 7b and c, for the two electrolytes investigated, when the HOPG is at open-circuit voltage, we observed unclear but still quasi-periodic zigzag features of EDL, where anions, cations and solvent molecules may coexist in each

layer. Alternate appearance of bright and dark layers when the voltage is applied indicates the formation of a hierarchical structure, as the positively polarized HOPG pushes the FSI⁻ anion toward the surface and repel the Li+ cation, which will induce the vertical separation of anions and cations, widening the interlayer spacing and eventually leading to the layer split. The phase diagram also shows that the distance of the closest layer to the surface plane increases with polarization. In addition, by monitoring the phase change of cantilever as a function of the distance between tip and HOPG surface, the density distribution of EDL on the interface is reflected in the phase-distance curve. The difference caused by NO₃⁻ is that more Li+ are retained in layered EDL after applying voltage, which makes a large part of solvent and anion bound by Li+ to form a polymer-like chains structure with higher viscosity, resulting in an obvious disparity in the density of EDL layer between the two electrolyte systems. As observed in phase-distance profiles, the degree of change in the phase transition of the EDL layer in the LiFSI–LiNO₃/DME system (3.982–11.32) after voltage application is significantly higher than that in the LiFSI/DME system (4.01–7.37), thus further confirming our hypothesis.

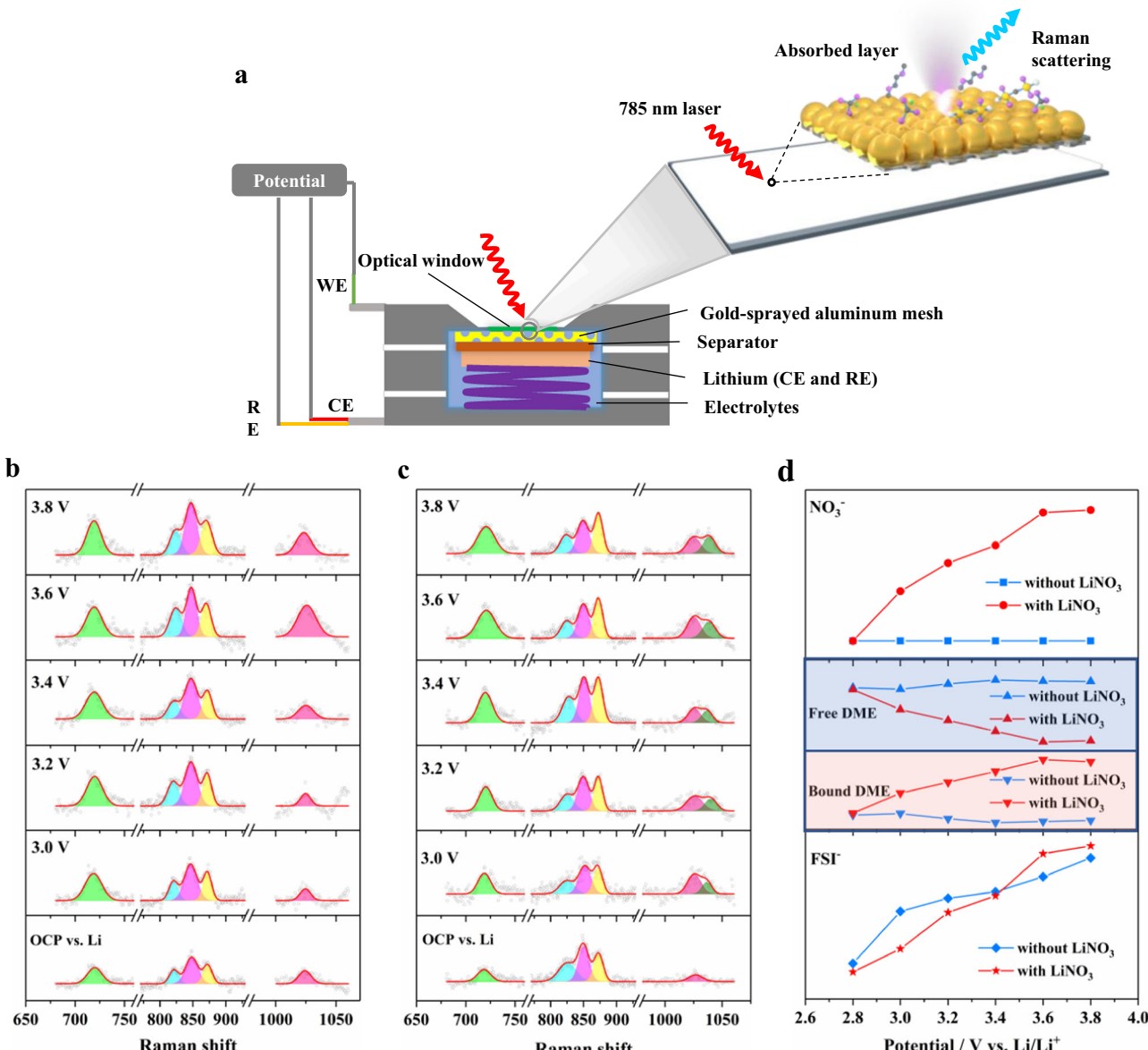

**Fig. 6 In situ EC-SERS characterization of EDL structure under different voltages. a** Schematic of an in situ EC-SERS cell, which was composed of a gold-sprayed aluminum mesh as a working electrode, a lithium foil as a reference electrode and counter electrode. Raman spectra of the surface layer of the working electrode under different voltages in (**b**) 1 M LiFSI/DME and (**c**) LiFSI–LiNO₃/DME. **d** The variation trend of each component (NO₃⁻, Free DME, Bound DME, FSI⁻) in the double layer of the two electrolytes with respect to voltage.

## Discussion

We have demonstrated here the intrinsic electrochemical properties of the electrolyte can be qualitatively changed by carefully engineering of a new kind of passivating and adaptive EDL structure at the electrode interface. Therefore, after EDL optimization, the lithium metal battery based on conventional concentration ($\leq 1$ M) of ether electrolyte exhibits record-high performances: a high cutoff voltage (as high as 4.5 V vs Li/Li$^+$), superior cycling performance under 5.0 C superfast charging/discharging rate, stable cycling life (>90% capacity retention over 600 cycles) and extremely low operating temperature of −91 °C (~42% of its room-temperature capacity) and stable charging/discharging cycling performance at −40 °C. This work not only provides an effective solution for improving the oxidation stability of ether-based electrolytes but also offers deep insights for understanding the role of the EDL in the electrolyte and could greatly expand the scope of electrolyte systems for next-generation high-voltage and ultra-low-temperature battery systems.

## Methods

**Materials**. Lithium nitrate (LiNO₃) (>99.99%), lithium perchlorate (>LiClO₄) (99.9%), lithium bis(trifluoromethanesulphonyl)imide (LiTFSI) (>99.9%), lithium bis(fluorosulfonyl)imide (LiFSI) (>99.9%), lithium tetrafluoroborate (LiBF₄) (>99.9%), and lithium triflate (LiTFO) (>99.9%), 1, 2-dimethoxyethane (DME) (>99.9%) and 1 m LiPF₆/EC + DEC (1:1 by volume) was purchased from Dodo-Chem. Tetrahydrofuran (THF) (>99.9%) was purchased from Shanghai Aladdin Bio-Chem Technology Co., LTD. By dissolving a predetermined amount of lithium salt (LiFSI) into a solvent of interest (DME or THF) and stirring, the electrolyte with a concentration of 1 mol L$^{-1}$ was prepared, and then 0.3 M of additives (LiNO₃, LiClO₄) was added to obtain the final electrolyte. Metallic Li foil was purchased by China Energy Lithium Co., LTD. Cathode NMC811 laminates were prepared by laying a mixture of 80 wt% NMC811 material, 10 wt% super-p and 10 wt% PVDF (8.0 wt% NMP) on an aluminum foil current collector. The high-loading density of Cathode NMC811 electrode (active mass loading: 21.0 mg cm$^{-2}$) was purchased from Guangdong Canrd New Energy Technology Co., LTD, and dried at 100 °C under vacuum before cell fabrication.

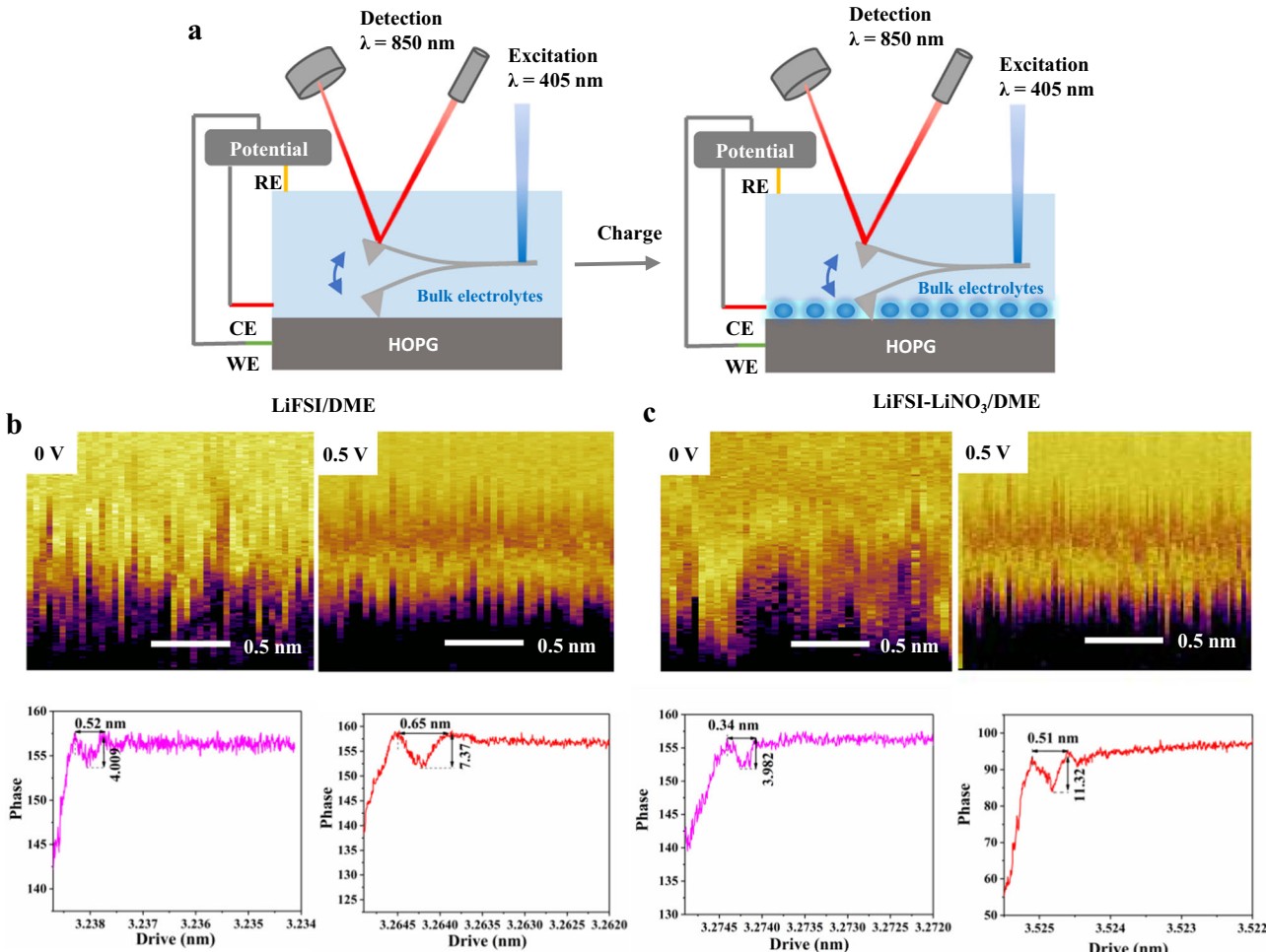

**Fig. 7 In situ EC-AFM characterization of EDL structure under different voltages. a** Schematic of an in situ EC-AFM cell, which was composed of a HOPG as working electrode, Ag/AgCl as reference electrode, and Pt wire as the counter electrode. The AFM cantilever is inside the liquid. Two-dimensional (2D)-AFM xz count maps HOPG–electrolytes interface at different electrode potential and corresponding Frequency shift versus distance curve in (**b**) 1 M LiFSI/DME and (**c**) LiFSI–LiNO₃/DME.

**Materials characterizations**. A JSM-7401F scanning electron microscopy (SEM) was conducted to characterize the surface morphology of lithium deposition. The Cu foil in a half cell after the deposition step was washed with 1,3-dioxolane (DOL) to remove residual lithium salts, dried, and then sealed in the glove box until transferred for characterization. The whole process of sample preparation is carried out in a glove box with oxygen and water contents below 0.1 ppm. The internal morphology of NMC811 particles was observed by SEM after being cut by a focused ion beam (FIB) at an accelerating voltage of 5 kV and a current of 86 pA. The chemical composition information of CEI was obtained by using X-ray photoelectron spectroscopy (XPS, ESCALAB Xi +) sputtering at different depths. The C1s peak at 284.6ev was used as the reference for all binding energies. The NMC811 electrode used for XPS testing was obtained by two cycles at a current density of 0.1 C and a current density of ten cycles at 0.5 C followed by cleaning with DOL solvent to remove the residual salt. The ionic conductivity of the electrolyte was measured using a standard 2032 coin cell with two polished 316 stainless steel electrodes placed symmetrically at a set distance. The electrolytic conductivity value is calculated by the following formula:

$$\sigma = \frac{L}{A \times R} \quad (1)$$

Where R is the resistance, and A and L are the area and spacing between the electrodes, respectively. Data points from 20 °C to −80 °C were measured using versastudio software and the symmetrical cells remained at a set temperature controlled by the thermostat for 1 h prior to the test.

**In situ surfaced-enhanced Raman spectroscopy (SERS)**. In situ surface-enhanced Raman spectroscopy (SERS) was performed in a sealed three-electrode electrochemical cell. A layer of coarse Au nanoparticles with a thickness of 20 nm was evaporated on the aluminum mesh as the working electrode with a surface enhancement effect and lithium as the reference electrode for the working

electrode. The electrochemical workstation (CHI 760e) was used to charge and discharge the three-electrode cell, and Raman scanning was carried out after the system reaches equilibrium for two minutes at each working potential (OCP vs Li,OCP + 0.2 V, OCP + 0.4 V, OCP + 0.6 V, OCP + 0.8 V). The Raman system (Horiba HR-800) used in this experiment was equipped with a 785 nm diode laser and a nominal power of 150 mW.

**In situ atomic force microscopy (AFM)**. Electrochemical two-dimensional atomic force microscopy (EC-2D-AFM) test obtained by Cypher VRS platform (Asylum Research, Oxford Instruments). The cantilever is immersed in a sealed three-electrode electrochemical cell, in which the working electrode is newly cut highly oriented pyrolytic graphite (HOPG), the counter electrode and the reference electrode are platinum and Ag/AgCl, respectively. The volume of electrolyte used in EC-2D-AFM measurement is generally in the range of 100–150 μL. After assembling the battery, we first carried out a fast large area scanning (at least 100 × 100 nm²) to confirm that the scanning area on the HOPG is atomically clean. The 2D force map is drawn by changing the X-coordinates measured by the force curve (Δf vs distance curve) to the next position and repeatedly measuring 128 times. All AFM experiments were carried out at room temperature.

**Electrochemical measurements**. In order to avoid the corrosion of the stainless steel by the electrolyte, the electrochemical comparison experiments of the LiFSI/DME and LiFSI–LiNO₃/DME systems both use an Al-clad cathode cases and an additional piece of aluminum foil underneath the cathode disk[44]. A detailed study on the corrosion behavior of stainless steel is placed in the Supplementary information. The charge–discharge performances of the Li||Cu, Li||Li, Li||NMC batteries were examined using 2032-type coin cells conducted on a battery test station (LANHE CT3001A). For low-temperature discharge experiments at different temperatures, the battery was charged at a current density of 0.5 C under room temperature and discharged at a current density 0.1 C under different

temperatures. The electrochemical stability windows of different electrolytes were tested in a half cell with Carbon coated aluminum foil as a working electrode, Li foil as the working electrode and reference electrode using CHI 760E electrochemical workstation. The assembled Li||NMC cells were rested at different temperatures (room temperature, $-20\,°C$, $-40\,°C$) for 2 h to achieve temperature equilibrium and then subjected to galvanostatic cycling. The Li||NMC811 half-cells is composed of 450 μm Li counter electrode and 0.7 mAh cm$^{-2}$ NMC811 electrode. The Li||NMC811 full cells utilized of 40 μm Li counter electrodes with 4.2 mAh cm$^{-2}$ NMC811 cathodes, N/P = 2.31. For low-temperature measurements, the electrochemical performance was measured in a calorstat (RK-7H-100LF, RIUKAI instrumenta or equipment CO.LTD) in the temperature range of 25 to $-40\,°C$. For the tests at $-74$, $-82$, and $-91\,°C$, the cells were immersed in solid/liquid mixtures of $CO_2$ (s)/methanol (L), $CO_2$ (s)/ethyl acetate (L), and $CO_2$ (s)/isopropanol (L), respectively, and where these mixtures were placed in a liquid nitrogen container.

**Computational methods**. Molecular dynamics (MD) simulations of the three lithium-ion electrolyte systems were performed using GROMACS[45] with the all-atom optimized potentials for liquid simulations (OPLS-AA) force field. The OPLS-2009IL force field parameters of FSI were obtained directly from literatures[46, 47] and a charge scaling of 0.8 was adopted to mimic polarization and charge transfer effects. The force field parameters of other organic molecules were generated using LigParGen web server[48]. The 1.2-scaling CM5 charges were based on DFT calculation by Gaussian program[49] and Hirshfeld population analysis by Multiwfn[50]. The force field parameters of graphene slabs were transferred from the default OPLS-AA force field in GROMACS and the atomic charge was corrected according to the DFT calculation of the surface electrostatic potential of the graphene surface with different net charges. The DFT calculations of graphene were performed using the Vienna Ab-Initio Simulation Package (VASP)[51, 52].

The initial simulation boxes of dimensions $51 \times 51 \times 300\,Å^3$ packed with electrolyte components were constructed using packmol program[53]. Their structures were first relaxed by energy minimization, and then underwent an annealing from 0 to 298.15 K with the time step of 1 ps during 1 ns to reach the equilibrium state. Velocity-rescale thermostat[54] with a relaxation constant of 1 ps was used to control the temperature at 298.15 K. Berendsen barostat with a isothermal compressibility constant of $4.5 \times 10^{-5}$ was used to control the pressure at $1.01325 \times 10^5$ Pa. Periodic boundary conditions were applied in all directions. Particle-mesh Ewald (PME) method with a cutoff distance of 10 Å was applied to treat the electrostatic interactions and the van der Waals forces. During the heat balance simulation of the NPT ensemble, the size of $x$ and $y$ axis were fixed to make sure the box size can fit the lattice constant of the graphene slab.

Upon quasi-equilibrium of the system, two graphene slabs with different net charges which represent the positive and negative electrodes are added. In order to reduce the Coulomb effect between the mirrored slabs due to the periodic boundary condition, a layer of the vacuum of 10 nm thickness was extended in the direction of z axis. After that, the energy minimization and heat equilibrium processes were all performed at the NPT ensemble with the same simulation parameters as above. An MD simulation for a total simulation time of 20 ns was then performed, and the trajectory was saved every 1 ps. The further statistics results were analyzed from the trajectory data by GROMACS tool-suites, visual molecular dynamics (VMD)[55], and some python scripts written by ourselves. The cluster search was performed in the inner layer of the Helmholtz electric double inner layer, and the main structures of solvation clusters and free molecules were DME, FSI, Li-(DME)$_2$, Li-DME-FSI, Li-(DME)$_2$-FSI, Li-(DME)$_3$, Li-DME-NO$_3$, and Li-(DME)$_2$-NO$_3$. All these clusters were then optimized under the framework of the density of functional theory (DFT) with PBE0 functional and def2-SVP basis set[56]. The frontier orbits and electrostatic surface potential (ESP) of these clusters were rendered using VMD and Gauss View programs, respectively. The electrostatic potential range corresponding to the color scale is from $-0.1$ to $+0.1$ a.u., the isosurface of the HOMO and LUMO orbits is 0.02 a.u. BF$_4^-$ and SO$_4^{2-}$ for comparison, in order to support, from a "negative" viewpoint, the design principle.

Ab initio molecular dynamics (AIMD) based on quantum chemical methods is used to further verify the reliability of the simulation results (Supplementary Figs. 26 and 27). AIMD simulation was performed via CP2K program. AIMD using the frozen-core, all-electron projector augmented wave (PAW) method and the generalized gradient approximation of Perdew, Burke, and Ernzerhof with the D2 correction for dispersion as parametrized by Grimme. Gamma-point computations with an energy cutoff of 400 eV, an electronic energy convergence criterion of $1 \times 10^{-4}$ eV, and a time step of 1.0 fs were used for ab initio molecular dynamics simulations.

Note that in our simulation system we adopted the constant charge method to mimic the electrode potential to form EDLs. Indeed, the constant potential method (CPM) is a more accurate but more computationally expansive method to model the applied potential[57–59]. However, the constant charge method (CCM) is generally considered to be able to simulate open electrode system at the equilibrium state (where the electrode surface is in contact with the bulk electrolyte[60], for example, a planar[57], cylindrical[61], or spherical[62] surface), while it may not be accurate to simulate the charging dynamics or nanoporous electrode system. Very recent work reported the first MD modeling of CPM simulations for exploring the galvanostatic charge–discharge of supercapacitors[31], revealing that

for the simulation system with the same electrolyte-electrode setup with ours in this work (i.e., electrolytes with the graphene slab electrodes), CPM and CCM give nearly the same EDL structure at any time during the charging and discharging process, although showing a significant deviation for nanoporous electrode systems.

## Data availability

The data that support the plots within this paper and another finding of this study are available from the corresponding author upon reasonable request.

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

## Acknowledgements

The authors thank Prof. Baoguo Wang for the helpful discussions. This research was financially supported by the National Nature Science Fund of China (22071133, K.L.), Tsinghua University Initiative Scientific Research Program (K.L.), and the National Key Research and Development Program (2019YFC0810703, K.L.).

## Author contributions

W.Z. and K.L. conceived the idea and designed the experiments. W.Z. prepared materials, performed measurements, and analyzed the data. L.W. helped with part of the schematic design. Y.L., P.Z., Y.X., S.Y., X.C., H.D., and H.Z. helped with part of the experiment and data analysis. All authors discussed the results and commented on the manuscript. W.Z. wrote the draft, K.L. revised and finalized the manuscript.

## Competing interests

The authors declare no competing interests.
