## [Peer Review File · Nature Communications]

Engineering a passivating electric double layer for high performance lithium metal batteriesREVIEWER COMMENTS

Reviewer #1 (Remarks to the Author):

In this contribution, Liu et al aims to enable fast charging and good low temperature performance in a LIB by “engineering” an EDL composed of tightly bounds Li⁺ ions embedded in what is purported to be a polymer like structure on the cathode. This is an interesting approach, since the hypothesis is that relatively free Li⁺ ions in the EDL can have deleterious effects on battery performance. The authors’ hope is that by creating what is essentially as pseudo-capacitor battery, they can mitigate so of these effects. While this work is rather expansive in its execution, this reviewer is not convinced by the central premise. Specifically, although the results clearly demonstrate that the stabilizing effect of Li⁺ produces an improved resistance towards oxidative decomposition in the system, there is little evidence to describe the improved rate performance and temperature scaling behavior. Furthermore, the choice of electrolyte chemistries for comparison makes the assessment of the authors’ EDL hypothesis impossible due to the number of confounding variables. The data presented here offers little evidence as to the mechanism of the EDL-provided performance improvement, which must be improved before this paper is publishable in Nat Comm:

- The improved oxidative stability of various ions in ether electrolytes is known to be highly dependent on choice of current collector/cathode cap material. For example, Ren et al. (Ren, X. et al. Enabling High-Voltage Lithium-Metal Batteries under Practical Conditions. *Joule* 2019, 3 (7), 1662–1676.) demonstrated that ~ 1 M LiFSI DME could cycle NMC811||Li cells for hundreds of cycles simply by applying an Aluminum coated cathode cap in coin cells. It is crucial that this strategy be implemented to compare with the LiFSI-LiNO₃-DME electrolyte to assess the validity of the EDL hypothesis.
- The LiFSI-LiNO₃-DME system is compared to 1 M LiPF₆ – EC/DEC, which does not have a single chemical component in common. To demonstrate the effect of the LiNO₃-induced EDL structure on the performance it is highly recommended to compare the LiFSI-LiNO₃-DME performance to 1 M LiFSI DME as well, which does not share the same EDL structure. The aforementioned Al protection on the cathode side of the coin cell should enable this. If the author’s EDL hypothesis is indeed valid, significant performance differences should be seen when comparing LiFSI-LiNO₃-DME and LiFSI-DME as well.
- What evidence do the authors have that the improved fast-charging performance observed in the LiFSI-LiNO₃-DME system is a result of the proposed cathode reactions as opposed to improved impedance on the anode side? 1 M LiPF₆ EC/DEC is well known to have extremely poor reversibility and rate performance on the Li metal side that may lead to a very high overpotential at high rate that may be driving the performance differences observed at high rate and low temperature. This is particularly worrying given that the author’s EDL hypothesis essentially only considers the cathode side. Therefore, 3-electrode experiments need to be conducted to deconvolute the anode and cathode contributions to the overpotential under various conditions.
- The notion that the improved viscosity/transport of ether electrolytes results in improve low temperature performance is highly disputed. A paper which you previously cited (Holoubek, J. et al. Tailoring Electrolyte Solvation for Li Metal Batteries Cycled at Ultra-Low Temperature. *Nature Energy* 2021, 6 (3), 303–313.) demonstrated that improved low-temperature performance was largely dependent on charge-transfer behavior, and not at all correlated to bulk ion transport. Other works have also found this to be the case (Li, Q. et al. Li⁺-Desolvation Dictating Lithium-Ion Battery’s Low-Temperature Performances. *ACS Appl. Mater. Interfaces* 2017, 9 (49), 42761–42768.; Zhang, S. S.; Xu, K.; Jow, T. R. The Low Temperature Performance of Li-Ion Batteries. *Journal of Power Sources* 2003, 115 (1), 137–140.). It is highly recommended that the impact of the altered EDL on the charge-transfer behavior be extensively discussed, provided that the comparison of LiFSI-DME and LiFSI-LiNO₃-DME yields significant performance improvements.
- If the authors still believe that the bulk transport in the electrolyte is indeed important to the high rate / low-temperature performance shown here, they must provide the ionic conductivity and transference number data for each electrolyte of interest.
- The authors claim that LiFSI-DME system without LiNO₃ cannot cycle NMC811||Li cells, but then immediately present XPS spectra of NMC 811 cathodes cycled 20 times in LiFSI-DME. This directly contradicts Supplementary Figure 3, which is supposedly run at the exact same rate as the cells shown for the XPS data.
- The authors hypothesize that a thinner CEI created by the LiFSI-LiNO₃-DME electrolyte (Figure

3) contributes to the improved rate performance shown in Figure 2. However, the performance compared in figure 2 are LiFSI-LiNO₃-DME and LiPF₆-EC-DEC, whereas the systems compared in Figure 3 are LiFSI-LiNO₃-DME and LiFSI-DME. This conflicting experimental design makes this hypothesis impossible to assess. It is therefore recommended once again to compare the NMC 811||Li cell performance between LiFSI-DME and LiFSI-LiNO₃-DME to support this hypothesis.

• Furthermore, to confirm that the CEI indeed contributes to the rate and low-temperature performance, the following experiment is recommended:

1. Cycle NMC 811 1 time in LiFSI-DME and LiFSI-LiNO₃-DME electrolytes to form the CEI.
2. Disassemble the cells and transfer the cathode into the opposite electrolyte and assess rate / low-temperature performance.

If the CEI is indeed rate limiting, the performance should correspond to the electrolyte the CEI is formed in instead of the final electrolyte applied for cycling

• The authors place a significant amount of weight on their MD studies, which supposedly verified their hypothesis. This is not entirely clear to me, however. Specifically, the authors used fixed charges for their atomic model (which was correctly scaled down in order to account for some polarization effects). However, this approach has only been validated in the bulk. How can we be sure that there aren't additional polarization effects near the interface and at the EDL? Application of a polarizable forcefield would go a long way towards settling this, or at the very least the authors should verify that the electrostatic potential of their current setup agrees with QM calculations.

• Furthermore, the use of fixed charges on the graphene surface to effect "voltage" is suspect [J. Chem. Phys. 141, 184102 (2014)], and can lead to spurious physics. The application of that approach here sows doubt on the central premise: the formation of polymer like layers near the cathode. The authors are encouraged to verify that these motifs survives a more rigorous theoretical examination (namely the constant potential method)

Reviewer #2 (Remarks to the Author):

This paper reports a novel approach that may enlarge the working voltage window toward higher voltage of ether-based electrolytes, such as DME, so that ether-based electrolytes can be used in the LIBs consisting of high-potential layered oxide cathodes and Li metal anode. In this novel approach, designed anions, such as NO₃⁻ and ClO₄⁻, are introduced as additives into ether-based electrolytes to modify the electric-double layer at the oxide cathode surfaces in order to effectively reduce the density of free, unbonded ether solvent molecules, resulting in substantially reduced solvent decomposition at high potential. The paper demonstrates that the successful employment of ether-based electrolytes open up the opportunities for extremely fast charging/discharging and ultra-low temperature applications of high-voltage LIBs. In addition, the paper provides both theoretical calculation and sophisticated surface sensitive analytic data to elucidating the underlying mechanism. The simplicity of the new method, the new application potential, and interesting scientific insight could render this paper having a broad impact in the LIB field. There are a few issues needed to be properly addressed before the paper can be recommended for publication.

1. The key point of this paper is the interfacial stability enhancement of the ether-based electrolyte in contact of the high-potential oxide cathode. This paper presents many high-rate data, but they often overlook the interfacial issue due to insufficient reaction time. The authors are suggested to compare the interfacial stability in a quantitative manner among different electrolyte systems. In one method, for example, the authors can run leakage current measurement at selected floating potentials (for example, at 4.3V) for DME electrolytes either with or without NO₃⁻ and a carbonate electrolyte, which can serve as a benchmark.
2. Also for the same reason, the charge-discharge voltage plots of formation cycle and other cycles of low C-rate (for example, 0.1C) should be provided for comparison.
3. In Fig. 2, the paper shows superior cycle stability of NO₃-added DME electrolyte to a carbonate electrolyte. It will be desirable to know if the difference originates solely from the Li metal side and how much the cathode side could contribute, either positively or negatively, to it? The authors may consider to run testing on oxide||oxide symmetric cells to remove the Li metal anode effects.

4. For theoretical calculation, calculation should also be carried out on other “non-improving” anion(s), such as SO_4^{2-} and CO_3^{2-} (Fig. SI 4) for comparison, in order to support, from a “negative” viewpoint, the design principle.

Other editorial points:

1. Symbols and legends in all figures should be enlarged.
2. The coulombic efficiency scale should be enlarged in all related figures.
3. The color scale in Fig. 4a and b should be explained in caption.
4. Figures 4g and h are very hard to read. Separate blown-up images showing much greater details (such as atom arrangement and bondings) should be given.

Reviewer #3 (Remarks to the Author):

The authors developed a new electrolyte with stable performance at high voltage and low temperature which is highly desirable for practical applications. They also explained this improved performance based on a stable EDL formed in cathode. The results are very impressive, and the manuscript is well written. Therefore, I recommend publication of this work in NC after addressing the following concerns.

1. One of the most critical properties of Li metal batteries is the coulombic efficiency of electrolyte tested in $\text{Li}|\text{Cu}$ cells. what is the CE of new electrolytes investigated in this work?
2. This work used stainless steel coin cell case for electrochemical test. It is known that low concentration LiFSI electrolyte is not stable with stainless steel coin cell case at a voltage higher than 4.2V. are there any corrosion effect observed on the stainless steel coin cell case used in this work? If not, what is the mechanism to avoid the corrosion at low LiFSI concentration?
3. The author should add the scale bar in the SEM image (Supplementary Fig.5).
4. In the experimental method, the author mentioned 30 μm of Li metal used for full cell, while in the caption of Fig. 2 and the line of 135 on page 7, the author said the 40 μm of Li used. What is the truth?
5. The ionic conductivities for various temperatures of electrolytes samples should be provided.
6. How is the morphologies of Li metal after cycles? The SEM images after cycling will be very helpful for the readers to have a better understanding on the

We thank all the reviewers for their valuable comments on our manuscript. Their constructive suggestions for improvement have certainly raised the quality of our manuscript. In order to address the reviewers concerns, we have addressed their comments point by point.

RESPONSE TO REVIEWERS' COMMENTS

REVIEWER 1:

In this contribution, Liu et al aims to enable fast charging and good low temperature performance in a LIB by “engineering” an EDL composed of tightly bounds Li⁺ ions embedded in what is purported to be a polymer like structure on the cathode. This is an interesting approach, since the hypothesis is that relatively free Li⁺ ions in the EDL can have deleterious effects on battery performance. The authors' hope is that by creating what is essentially as pseudo-capacitor battery, they can mitigate so of these effects. While this work is rather expansive in its execution, this reviewer is not convinced by the central premise.

Specifically, although the results clearly demonstrate that the stabilizing effect of Li⁺ produces an improved resistance towards oxidative decomposition in the system, there is little evidence to describe the improved rate performance and temperature scaling behavior.

Furthermore, the choice of electrolyte chemistries for comparison makes the assessment of the authors' EDL hypothesis impossible due to the number of confounding variables.

The data presented here offers little evidence as to the mechanism of the EDL-provided performance improvement, which must be improved before this paper is publishable in Nat Comm:

Response: Thank you very much for your many important and valuable suggestions, and we have added extensive new experiments and theoretical simulations to fully confirm our hypothesis. We hope the revised manuscript is now suitable for publication in Nature Communications.

Comment 1: *The improved oxidative stability of various ions in ether electrolytes is known to be highly dependent on choice of current collector/cathode cap material. For example, Ren et al. (Ren, X. et al. Enabling High-Voltage Lithium-Metal Batteries under Practical Conditions. Joule 2019, 3 (7), 1662–1676.) demonstrated that ~ 1 M LiFSI DME could cycle NMC811//Li cells for hundreds of cycles simply by applying an Aluminum coated cathode cap in coin cells. It is crucial that this strategy be implemented to compare with the LiFSI-LiNO₃-DME electrolyte to assess the validity of the EDL hypothesis.*

Response to comment 1: We gratefully appreciate for your valuable suggestion. We followed your advice and re-compared the electrochemical behavior of LiFSI/DME and LiFSI-LiNO₃/DME by applying an Aluminum coated cathode cap in coin cells. The following new experiments were carried out during the revision.

(1) Firstly, both rate and cycling tests ranging from low rate (0.1 C) to high rate (10.0 C) were conducted for NMC811 electrodes in LiFSI/DME and LiFSI-LiNO₃/DME electrolytes. Indeed, 1M LiFSI/DME can be cycled in Li/NMC811 cell for several times after eliminating the factor of

stainless steel corrosion. However, As shown in new Fig. 2b in the revised manuscript, LiFSI/DME suffers rapid capacity fading with increasing current. On the contrary, the LiFSI-LiNO₃/DME exhibits a remarkable rate performance, achieving a discharge capacity of 152 mAh g⁻¹ even at a high rate of 10.0 C. Moreover, the cell can still maintain a stable cycle after returning from the rate of 10.0 C to the rate of 0.5 C, which proves the sustainable adaptability of LiFSI-LiNO₃/DME to the highly active NMC811 cathode material.

(2) The long cycle tests of LiFSI-LiNO₃/DME at different rates are extremely stable (new Fig.2c, new Supplementary Fig. 3-8), while LiFSI/DME is subject to continuous capacity decay at all rates, demonstrating the important role of LiNO₃ in improving the oxidation-resistance of DME-based electrolytes.

(3) The reactivity of the different electrolytes with NMC811 cathodes was further investigated by recording the open circuit voltage (OCV) of a fully charged cells as a function of storage time. As shown in new Fig. 2d, the OCV of a fully charged cells with LiFSI/DME drastically decreased after circuit disconnection, whereas a cell charged in LiFSI-LiNO₃/DME exhibited relatively little OCV change, even better than conventional commercial carbonate electrolytes.

(4) The new Fig. 2e shows the leakage current as a function of time generated by a potentiostatic hold (4.3V) for cells with different electrolytes. The initial decrease in the curve is the relaxation of the concentration gradient in the cell, and the final static leakage current value is usually used as an indicator to quantify the rate of the parasitic reaction. LiFSI/DME shows a much higher static leakage current than the other two electrolytes, indicating a faster parasitic reaction between electrode and electrolyte. The presence of NO₃⁻ significantly improves the stability of the electrode/electrolyte interface, thereby effectively diminishing reaction rate and inhibiting the self-discharge of a fully charged DME-based cells.

Thus, even by applying an Aluminum coated cathode cap, our designed strategy still works and show much better performance.

We have added these new data and corresponding discussions as Fig.2, Supplementary Fig. 2-8 on page 6-8 in the revised main text.

Fig. 2. Comparison of electrochemical properties of LiFSI/DME and LiFSI-LiNO₃/DME. (a) Oxidative stability measured via LSV for Li||Al cells. (b) Rate capability of Li||NMC811 cells under different charging/discharging rates. (c) Long-term cycle performance of Li||NMC811 cells using different electrolytes at 1.0 C rate. (d) Self-discharge tests after a potentiostatic hold at 4.3 V vs Li/Li⁺. (e) Typical current relaxation curves collected from NMC811/Li half-cells during a potentiostatic hold at 4.3 V vs Li/Li⁺.

Supplementary Fig. 3. Long-term cycling performance of Li||NMC811 cells using LiFSI/DME and LiFSI-LiNO₃/DME at 0.1 C rate.

Supplementary Fig. 4. Long-term cycling performance of Li||NMC811 cells using LiFSI/DME and LiFSI-LiNO₃/DME at 1.0 C rate.

Supplementary Fig. 5. Long-term cycling performance of Li||NMC811 cells using LiFSI/DME and LiFSI-LiNO₃/DME at 3.0 C rate.

Supplementary Fig. 6. Long-term cycling performance of Li||NMC811 cells using LiFSI/DME and LiFSI-LiNO₃/DME at 5.0 C rate.

Supplementary Fig. 7. Long-term cycling performance of Li||NMC811 cells using LiFSI/DME and LiFSI-LiNO₃/DME at 10.0 C rate.

Supplementary Fig. 8. Long-term cycling performance of Li||NMC811 cells using LiFSI/DME and LiFSI-LiNO₃/DME at 5.0 C charge and 1.0 C discharge.

Comment 2: The LiFSI-LiNO₃-DME system is compared to 1 M LiPF₆-EC/DEC, which does not

have a single chemical component in common. To demonstrate the effect of the LiNO₃-induced EDL structure on the performance it is highly recommended to compare the LiFSI-LiNO₃-DME performance to 1 M LiFSI/DME as well, which does not share the same EDL structure. The aforementioned Al protection on the cathode side of the coin cell should enable this. If the author's EDL hypothesis is indeed valid, significant performance differences should be seen when comparing LiFSI-LiNO₃-DME and LiFSI-DME as well.

Response to comment 2: We are sorry that our comparative experiment may have caused you some confusion, and we have also made some adjustments to the structure of this article. As you suggested, we complement a more adequate comparative test between LiFSI-LiNO₃/DME and LiFSI-DME to demonstrate the high electrochemical stability of the LiNO₃-derived EDL structure as well as the significant improvement in the electrochemical properties, as can be seen from the Response to your comments 1 and the following comments.

Logically next, since the problem of ether solvent decomposition at high voltage is effectively solved (one of the biggest obstacle for its application), ether-based electrolyte that can sustain high voltage can do things that carbonate-based electrolyte cannot do, such as fast charging and ultra-low temperature operation, because ether-based electrolyte has significantly higher conductivity, higher transference number and wider liquid range. But, in this comparison part, we do not want to take the change of EDL into account, because as you pointed out, no chemical components of the two electrolyte systems are the same.

Thus, to avoid confusion, we have revised the corresponding parts in the revised main text as follows.

Line 99-248 in revised main text:

Electrochemical performance and characterization of the electrolyte. Firstly, we verify whether the two candidates can improve the cathodic stability of ether-based electrolyte, as presented in Fig. 2a, the degradation potential (take 0.05 mA cm⁻² as the criterion) of 1 M Lithium bis(fluorosulfonyl)imide/1,2-Dimethoxyethane (LiFSI/DME) is earlier than 4.0 V (vs Li/Li⁺), while the presence of 0.3 M LiNO₃ broadens the electrochemical stability window to 4.3 V, and the decomposition potential is further widened to 4.5 V with the addition of 0.3 M LiClO₄, which is even more significant than the effect of the means of high concentration electrolytes (4.2 V vs Li/Li⁺ for 4.0 M LiFSI/DME). Next, we take 0.3 M LiNO₃ additive as an example to investigate the compatibility of LiFSI-LiNO₃/DME with high voltage cathode materials (NMC811) and verify the proposed voltage responsive mechanism. **Both rate and cycling tests ranging from low rate (0.1 C) to high rate (10.0 C) were conducted for NMC811 electrodes in LiFSI/DME and LiFSI-LiNO₃/DME electrolytes. As shown in Fig. 2b, as expected, LiFSI/DME suffers rapid**

capacity fading with increasing current. On the contrary, the LiFSI-LiNO₃/DME exhibits a remarkable rate performance, achieving a discharge capacity of 152 mAh g⁻¹ even at a high rate of 10.0 C. Moreover, the cell can still maintain a stable cycle after returning from the rate of 10.0 C to the rate of 0.5 C, which proves the sustainable adaptability of LiFSI-LiNO₃/DME to the highly active NMC811 cathode material. It is worth noting that the coulombic efficiency of LiFSI/DME in formation cycle is only ~ 68.44 %, while that of LiFSI-LiNO₃/DME in the formation cycle is significantly increased to 85.98 % (Supplementary Fig. 2). In addition, the long cycle tests of LiFSI-LiNO₃/DME at different rates are extremely stable (Fig.2c, Supplementary Fig. 3-8), while LiFSI/DME is subject to continuous capacity at all rates, demonstrating the important role of LiNO₃ in improving oxidation-resistance of DME-based electrolytes. Some other additives have also proved to be ineffective in improving cycle stability (Supplementary Fig. 9). The reactivity of the different electrolytes with NMC811 cathodes was further investigated by recording the open circuit voltage (OCV) of a fully charged cells as a function of storage time. As shown in Fig. 2d, the OCV of a fully charged cells with LiFSI/DME drastically decreased after circuit disconnection, whereas a cell charged in LiFSI-LiNO₃/DME exhibited relatively little OCV change, even better than conventional commercial electrolytes. Fig. 2e shows the leakage current as a function of time generated by a potentiostatic hold (4.3 V) for cells with different electrolytes. The initial decrease in the curve is the relaxation of the concentration gradient in the cell, and the final static leakage current value is usually used as an indicator to quantify the rate of the parasitic reaction. Obviously, LiFSI/DME shows a much higher static leakage current than the other two electrolytes, indicating a faster parasitic reaction between electrode and electrolyte. The presence of NO₃⁻ significantly improves the stability of the electrode/electrolyte interface, thereby effectively diminishing reaction rate and inhibiting the self-discharge of a fully charged DME-based cells. The extensive intergranular cracking in the NMC811 cathode cycled with LiFSI/DME (Supplementary Fig. 10a) means more side reactions as well as faster electrolyte consumption that will eventually result in a cliff-like decay of capacity. However, the electrode particles cycled in LiFSI-LiNO₃/DME system preserved mechanical integrity without visible cracks (Supplementary Fig. 10b), further supporting superior compatibility and stability of this electrode/electrolyte interphase.

Fig. 2. Comparison of electrochemical properties of LiFSI/DME and LiFSI-LiNO₃/DME. (a) Oxidative stability measured via LSV for Li||Al cells. (b) Rate capability of Li||NMC811 cells under different charging/discharging rates. (c) Long-term cycle performance of Li||NMC811 cells using different electrolytes at 1.0 C rate. (d) Self-discharge tests after a potentiostatic hold at 4.3 V vs Li/Li⁺. (e) Typical current relaxation curves collected from NMC811/Li half-cells during a potentiostatic hold at 4.3 V vs Li/Li⁺.

Moreover, compared to commercial carbonate electrolytes, DME-based electrolytes exhibit significantly higher conductivity at all tested temperatures, as well as higher Li⁺ transference number. As shown in Fig.3a, the conductivity of the DME-based electrolyte is observed to retain a

remarkable value (3.68 mS cm^{-1} for LiFSI/DME, 2.87 mS cm^{-1} for LiFSI-LiNO₃/DME) even at $-60 \text{ }^\circ\text{C}$ are attributed to the low melting point ($\sim -69 \text{ }^\circ\text{C}$), compared with the values less than 0.1 mS cm^{-1} in LiPF₆/EC+DEC at temperatures below $-20 \text{ }^\circ\text{C}$. The Li⁺ transference number of DME-based electrolyte reported in Fig.3b all exceed 0.4 (0.45 for LiFSI/DME, 0.43 for LiFSI-LiNO₃/DME), while that of commercial carbonate-based electrolytes is only 0.31. Therefore, after solving the problem of poor oxidation stability of DME-based electrolyte, the high voltage lithium metal battery with DME-based electrolyte and NMC811 cathode can be expected to achieve faster charge/discharge rates and lower operating temperatures while obtaining high energy density at the same time. As shown in Fig. 3c, the LiFSI-LiNO₃/DME system maintained a high discharge capacity of 150 mAh g^{-1} (69.7% of the capacity at 0.2 C rate) at 10.0 C rate, whereas the capacity of the 1M LiPF₆/EC+DEC system was limited to only 30 mAh g^{-1} (15% of the capacity at 0.2 C rate). The three-electrode experiments are implemented to decouple the anode and cathode contributions to the overpotential at different rates. As shown in Supplementary Fig. 11, the overpotential of lithium metal anode in ether-based electrolytes is much smaller than that of carbonate-based electrolytes at low rates (0.2 C, 0.5 C) due to the superior compatibility between lithium metals and ether solvents, but there is no significant difference in the overpotential of the two electrolytes on the cathode side. However, with the increase of the charge/discharge rate, the struggling dynamic behavior of the carbonate-based electrolyte on the cathode side dominates the performance loss. Therefore, the excellent fast charging performance of Li/NMC811 cells with LiFSI-LiNO₃/DME is mainly attributed to the fast kinetic response of our electrolyte on the cathode side. Moreover, Fig. 3d and 3e demonstrated the sustainable super-fast charging/discharging capability of the LiFSI-LiNO₃/DME system, which maintains 90% capacity after 700 cycles at 5.0 C charge/discharge rate, while the cell with commercial carbonate electrolyte systems suffered from severe capacity decay accompanied by unstable coulombic efficiency. It is well known that the LiNO₃ additive could make the electrochemical behavior of Li metal anode even better (Supplementary Fig. 12). Thus, the high-voltage Li||NMC811 full cells (thin Li anode: $40 \text{ }\mu\text{m}$, high-loading cathode NMC811: 20.3 mg cm^{-2} , the N/P ratios of the Li||NMC811 cell was 2.31) were tested to further explore utility function of the LiFSI-LiNO₃/DME system under highly challenging conditions. As shown in Fig. 3f, Li||NMC811 full cells using LiFSI-LiNO₃/DME system achieved absolute superior cycling performance even at a high rate of 2.0 C with 82% capacity retention after 200 cycles (Fig. 3g). To verify the universality of the proposed mechanism discussed above, we prove that the LiFSI-LiNO₃/DME system can successfully pair with another high-voltage cathode LiCoO₂ (Supplementary Fig. 13a, b). Further, we replace the 0.3 M LiNO₃ with 0.3 M LiClO₄, found that the LiFSI-LiClO₄/DME system can also pair NMC811 cathodes and demonstrate superior cycling

performance at high cutoff voltage of 4.5 V (Supplementary Fig. 13c, d). In addition, we also confirmed that (LiFSI-LiNO₃/Tetrahydrofuran) LiFSI-LiNO₃/THF system can be successfully applied to Li||NMC811 cells by replacing DME with THF (Supplementary Fig. 13e, f). Moreover, ether solvents generally exhibit extremely low freezing point and low viscosity, making it one of the most ideal candidates for low temperature battery. As shown in Supplementary Fig. 14a-14c, in contrast to the notorious poor low temperature performance of commercial electrolytes, LiFSI-LiNO₃/DME can be stably cycled 200 times at -20 °C. Extensive previous studies suggest that low-temperature performance depends largely on the charge transfer step, similarly demonstrated by our three-electrode experiments. Specifically, as shown in Supplementary Fig. 15e and f, the LiFSI/DME electrolyte shows significantly lowered charge transfer resistance than LiPF₆/EC+DEC electrolyte, and the LiFSI-LiNO₃/DME further lowered this value. The kinetics of different interfacial processes were further studied by temperature-dependent electrochemical impedance spectroscopy (EIS) (Supplementary Fig. 16). The activation energies of each interfacial process on the cathode side were obtained by fitting EIS spectra according to the classical Arrhenius law. LiFSI/DME shows a slightly reduced activation energy ($E_{a,ct}=50.21$ kJ mol⁻¹) compared to LiPF₆/EC+DEC ($E_{a,ct}=53.97$ kJ mol⁻¹). Interestingly, the addition of LiNO₃ further weakened the activation energy ($E_{a,ct}=44.23$ kJ mol⁻¹). Therefore, we believe that the better low-temperature performance of DME-based electrolyte than carbonate-based electrolyte is the result of the integrated contribution of bulk transport as well as interfacial transport. Moreover, the altered EDL due to NO₃⁻ even slightly accelerates the Li⁺ charge transfer behavior of DME-based electrolytes. More surprisingly, the Li||NMC811 cells using LiFSI-LiNO₃/THF can even operate at an extreme temperature of -91°C, delivering a capacity of 86 mAh g⁻¹ (Fig. 3i), attributing to extremely low freezing point of THF (-126.4 °C) and higher conductivity at ultra-low temperature (Supplementary Fig.17). As a result, in ultra-low temperature (-40 °C) long-term cycling, Li||NMC811 cells with LiFSI-LiNO₃/THF shows excellent stability, providing an initial capacity of 160 mAh g⁻¹ and 82% capacity retention after 150 cycles (Supplementary Fig. 14d). Further, the LiFSI-LiNO₃/THF-based full cells was cycled at -40° to provide the holistic effect of both the lithium metal anode and the NMC811 cathode at ultra-low temperature. As displayed in Fig. 3h, stable battery cycle in full cells can be achieved using LiFSI-LiNO₃/THF electrolytes, retaining 68% of the initial areal capacity after 80 cycles, demonstrating the compatibility of the electrolyte system with the cathode and anode and the reversibility of the electrochemical process at ultra-low temperature. As far as we know, there is no report so far that a lithium metal battery system can combine high cut-off voltage, fast charging capability and ultra-low temperature performance (Supplementary Fig.18, Supplementary Table. 3).

Fig. 3. Comparison of transport and electrochemical properties between ether-based electrolyte and carbonate-based electrolyte. (a) Conductivity versus temperature. **(b)** Li^+ transference number (τ_{Li^+}) computed from DC polarization measurements at 10 mV using the Bruce–Vincent method. **(c)** Rate capability of Li||NMC811 cells under different charging/discharging rates. **(d)** Long-term super-fast charging/discharging performance of Li||NMC811 cells using different electrolytes at 5.0 C rate. **(e)** the corresponding voltage profiles at different cycles of Li||NMC811 cells using LiFSI-LiNO₃/DME at 5.0 C rate. **(f)** Long-term cycling performances of high-voltage Li||NMC811 full batteries with 40 μm Li anode. The N/P ratios of the Li||NMC811 cell was 2.31. The first two formation cycles were carried out at a 0.1 C rate, followed by 15 cycles at 0.5 C rate, sequential 15 cycles at 1.0 C rate and the long-term cycling was at 2.0 C rate. **(g)** The corresponding voltage profiles of high-voltage Li||NMC811 full batteries using LiFSI-LiNO₃/DME. **(h)** Cycling performance of full cells in LiFSI-LiNO₃/THF electrolyte at -40 $^{\circ}\text{C}$ and 0.3 C rate. **(i)** Discharge profiles of Li||NMC811 cells using

LiFSI-LiNO₃/THF electrolyte at different temperatures.

Comment 3: *What evidence do the authors have that the improved fast-charging performance observed in the LiFSI-LiNO₃-DME system is a result of the proposed cathode reactions as opposed to improved impedance on the anode side? 1 M LiPF₆ EC/DEC is well known to have extremely poor reversibility and rate performance on the Li metal side that may lead to a very high overpotential at high rate that may be driving the performance differences observed at high rate and low temperature. This is particularly worrying given that the author's EDL hypothesis essentially only considers the cathode side. Therefore, 3-electrode experiments need to be conducted to deconvolute the anode and cathode contributions to the overpotential under various conditions.*

Response to comment 3: Thank you very much for your valuable suggestions. As you suggested, the three-electrode experiments are implemented to decouple the anode and cathode contributions to the overpotential at different rates.

As shown in new Supplementary Fig. 11, the overpotential of lithium metal anode in ether-based electrolytes is much smaller than that of carbonate-based electrolytes at low rates (0.2 C, 0.5 C) due to the superior compatibility between lithium metals and ether solvents, but there is no significant difference in the overpotential of the two electrolytes on the cathode side. However, with the increase of the charge/discharge rate, the struggling dynamic behavior of the carbonate-based electrolyte on the cathode side dominates the performance loss. Therefore, the excellent fast charging performance of Li/NMC811 cells with LiFSI-LiNO₃/DME is mainly attributed to the fast kinetic response of our electrolyte on the cathode side.

We have added these new data as new Supplementary Fig. 11 in SI, and the corresponding explanations were added on lines 169-179 on page 9-10 in the revised manuscript.

Supplementary Fig. 11. Voltage curves of the lithium metal anodes (right axis) and NMC811 cathodes (Left axis) with respect to the lithium foil reference electrodes during charge and discharge at (a) 0.2 C (b) 0.5 C (c) 1.0 C (d) 3.0 C (e) 5.0 C current density.

Comment 4: The notion that the improved viscosity/transport of ether electrolytes results in improve low temperature performance is highly disputed. A paper which you previously cited (Holoubek, J. et al. Tailoring Electrolyte Solvation for Li Metal Batteries Cycled at Ultra-Low Temperature. *Nature Energy* 2021, 6 (3), 303–313.) demonstrated that improved low-temperature performance was largely dependent on charge-transfer behavior, and not at all correlated to bulk ion transport. Other works have also found this to be the case (Li, Q. et al. Li⁺- Desolvation Dictating Lithium-Ion Battery's Low-Temperature Performances. *ACS Appl. Mater. Interfaces* 2017, 9 (49), 42761–42768.; Zhang, S. S.; Xu, K.; Jow, T. R. The Low Temperature Performance of Li-Ion Batteries. *Journal of Power Sources* 2003, 115 (1), 137–140.). It is highly recommended that the impact of the altered EDL on the charge-transfer behavior be extensively discussed,

provided that the comparison of LiFSI-DME and LiFSI-LiNO₃-DME yields significant performance improvements.

Response to comment 4: Thank you very much for your valuable suggestions. The improved low-temperature performance caused by the increased viscosity/transport of ether-based electrolyte mentioned in this paper is based on the carbonate-based electrolyte as a control, which does not take into account the change of EDL. But, you mentioned the impact of the changed EDL on fast charge performance and low temperature performance is a very meaningful point of view, which is also of great interest to us. Therefore, we did the following measurements during the revision.

(1) Kinetics of interfacial processes at the cathode/electrolyte interface were measured by EIS using a 3-electrode setup (Supplementary Fig. 16). The activation energies of each interfacial process on the cathode side were obtained by fitting EIS spectra according to the classical Arrhenius law. LiFSI/DME shows a slightly reduced activation energy ($E_{a,ct}=50.21 \text{ kJ mol}^{-1}$) compared to LiPF₆/EC+DEC ($E_{a,ct}=53.97 \text{ kJ mol}^{-1}$). Interestingly, the addition of LiNO₃ further weakened the activation energy ($E_{a,ct}=44.23 \text{ kJ mol}^{-1}$). Therefore, the altered EDL significantly facilitated the charge-transfer behavior at the interface.

(2) We supplemented 3-Electrode experiments to test the charge transfer behavior at low temperatures on the positive side, as shown in Supplementary Fig. 15. Indeed, as you pointed out, the result shows that the low-temperature performance depends largely on the charge transfer step. Specifically, as shown in Supplementary Fig. 15e and f, the LiFSI/DME electrolyte shows significantly lowered charge transfer resistance than carbonate electrolyte, and the LiFSI-LiNO₃/DME further lowered this value. Thus, the altered EDL facilitated the charge-transfer behavior at the cathode/electrolyte interface.

We have added these new data as new Supplementary Figs. 15 and 16 in SI, and the corresponding explanations were added on lines 203-218 on page 11 in the revised manuscript.

Supplementary Fig. 16. Kinetics of interfacial processes at the cathode/electrolyte interface measured by EIS using a 3-electrode setup. Temperature dependent EIS curves of cells containing (b) $\text{LiPF}_6/\text{EC}+\text{DEC}$ (c) LiFSI/DME and (d) $\text{LiFSI-LiNO}_3/\text{DME}$ and WSE. (e) Arrhenius behavior of the resistance.

Supplementary Fig. 15. 3-Electrode impedance study of NMC811||Li||Li cells at 25 °C and -20 °C a) Working schematic of the 3-electrode cell and Equivalent circuit model. Impedance spectra of the cathode at 50% state-of-charge (SOC) in b) $LiPF_6/EC+DEC$, c) $LiFSI/DME$ and d) $LiFSI-LiNO_3/DME$. Breakdown of equivalent circuit elements from fit at (e) 25 °C and (f) -20 °C.

Comment 5: If the authors still believe that the bulk transport in the electrolyte is indeed important to the high rate/low-temperature performance shown here, they must provide the ionic conductivity and transference number data for each electrolyte of interest.

Response to comment 5: Thank you very much for your valuable suggestions. We have supplemented the ion conductivity at different temperatures and the Li^+ transference number data for each electrolyte system of interest. As shown in new Figure 3, Supplementary Fig. 17 and Supplementary table 2.

(1) Compared to commercial carbonate electrolytes, ether-based electrolytes exhibit significantly higher conductivity at all tested temperatures, as well as higher Li^+ transference number. As shown in Fig.3a, at -60 °C, the conductivity of the $LiFSI/DME$ electrolyte is 3.68 mS cm^{-1} and the Li^+ transference number is 0.45, and the values of $LiFSI-LiNO_3/DME$ electrolyte is 2.87 mS cm^{-1} and 0.43. The conductivity of the THF-based electrolyte showed a more significant advantage at ultralow temperature. Even at -80 °C, the conductivity of the $LiFSI/THF$ electrolyte is 2.06 mS cm^{-1} .

cm⁻¹ and the transference number is 0.49, and the values of LFSI/LiNO₃/THF electrolyte is 1.22 mS cm⁻¹ and 0.46). In great contrast, in LiPF₆/EC+DEC, the conductivity is below 0.1 mS cm⁻¹ even at temperatures below -20 °C, with a low transference number of 0.31. It is noteworthy that only slight variation in the conductivity and Li⁺ transference number of the DME-based electrolytes were observed due to the presence of LiNO₃.

(2) We also calculated the charge-transfer energy barrier of ether-based electrolytes and carbonate-based electrolytes by using temperature-dependent electrochemical impedance spectroscopy (EIS). LiFSI/DME shows a slightly reduced activation energy ($E_{a,ct}=50.21$ kJ mol⁻¹) compared to LiPF₆/EC+DEC ($E_{a,ct}=53.97$ kJ mol⁻¹). Interestingly, the addition of LiNO₃ further weakened the activation energy ($E_{a,ct}=44.23$ kJ mol⁻¹).

(3) We supplemented 3-Electrode experiments to test the charge transfer behavior at low temperatures on the positive side, as shown in Supplementary Fig. 16. Indeed, as you pointed out, the result shows that the low-temperature performance depends largely on the charge transfer step. Specifically, as shown in Supplementary Fig. 16e and f, the LiFSI/DME electrolyte shows significantly lowered charge transfer resistance than LiPF₆/EC+DEC electrolyte, and the LiFSI-LiNO₃/DME further lowered this value.

Overall, it can be concluded that (1) the better fast charging and low temperature performance of ether-based electrolyte than carbonate electrolyte is the result of the integrated contribution of bulk transport as well as interfacial transport. (2) In ether electrolyte, the altered EDL due to NO₃⁻ accelerates the charge transfer behavior, which should be the main reason for the improved electrochemical performances of LiFSI-LiNO₃/DME.

We have added these new data as new Fig. 3, Supplementary Fig.15-17 and Supplementary table 2., the corresponding explanations were added on lines 153-165 on page 9, lines 203-218 on page 11 in the revised manuscript.

Fig. 3. Comparison of transport and electrochemical properties between ether-based electrolyte and carbonate-based electrolyte. (a) Conductivity versus temperature. (b) Li⁺ transference number (τ_{Li^+}) computed from DC polarization measurements at 10 mV using the Bruce–Vincent method. (c) Rate capability of Li||NMC811 cells under different charging/discharging rates. (d) Long-term super-fast charging/discharging performance of Li||NMC811 cells using different electrolytes at 5.0 C rate. (e) the corresponding voltage profiles at different cycles of Li||NMC811 cells using LiFSI-LiNO₃/DME at 5.0 C rate. (f) Long-term cycling performances of high-voltage Li||NMC811 full batteries with 40 μ m Li anode. The N/P ratios of the Li||NMC811 cell was 2.31. The first two formation cycles were carried out at a 0.1 C rate, followed by 15 cycles at 0.5 C rate, sequential 15 cycles at 1.0 C rate and the long-term cycling was at 2.0 C rate. (g) The corresponding voltage profiles of high-voltage Li||NMC811 full batteries using LiFSI-LiNO₃/DME. (h) Cycling performance of full cells in LiFSI-LiNO₃/THF electrolyte at -40 °C and 0.3 C rate. (i) Discharge profiles of Li||NMC811 cells using LiFSI-LiNO₃/THF electrolyte at different temperatures.

Supplementary Table 2. Li^+ transference number (τ_{Li^+}) computed from DC polarization measurements at 10 mV using the Bruce–Vincent method.

Electrolytes	τ_{Li^+}
$\text{LiPF}_6/\text{EC}+\text{DEC}$	0.31
LiFSI/DME	0.45
$\text{LiFSI}-\text{LiNO}_3/\text{DME}$	0.43
LiFSI/THF	0.49
$\text{LiFSI}-\text{LiNO}_3/\text{THF}$	0.46

Supplementary Fig. 17. Conductivity versus temperature of different electrolytes.

Supplementary Fig. 15. 3-Electrode impedance study of NMC811||Li||Li cells at 25 °C and -20 °C a) Working schematic of the 3-electrode cell and Equivalent circuit model. Impedance spectra of the cathode at 50% state-of-charge (SOC) in b) LiPF₆/EC+DEC, c) LiFSI/DME and d) LiFSI-LiNO₃/DME. Breakdown of equivalent circuit elements from fit at (e) 25 °C and (f) -20 °C.

Supplementary Fig. 16. Kinetics of interfacial processes at the cathode/electrolyte interface measured by EIS using a 3-electrode setup. Temperature dependent EIS curves of cells containing (b) LiPF₆/EC+DEC (c) LiFSI/DME and (d) LiFSI-LiNO₃/DME and WSE. (e) Arrhenius behavior of the resistance.

Comment 6: The authors claim that LiFSI-DME system without LiNO₃ cannot cycle NMC811||Li cells, but then immediately present XPS spectra of NMC 811 cathodes cycled 20 times in LiFSI-DME. This directly contradicts Supplementary Figure 3, which is supposedly run at the exact same rate as the cells shown for the XPS data.

Response to comment 6: Thank you so much for your careful check. We are very sorry for the confusion from our labeling error. But in the revised manuscript, we re-conducted the XPS measurements and compared the data of NMC811 cathodes cycled in LiFSI/DME and LiFSI-LiNO₃/DME at the exact same rate by applying an Aluminum coated cathode cap in coin

cells (as you suggested in Comment 1). Please see the specific analysis of new XPS data on pages 14-16 of the revised manuscript. In general, no significant component differences were found on the top surface of the CEI formed in the two electrolytes, except for slight variations in composition concentration and higher LiF content. However, after 50 cycles, the thickness of the CEI layer formed in LiFSI/DME increased dramatically (5nm to 20nm), indicating that the initial formed CEI could not effectively protect the electrode/electrolyte interface. In contrast, the LiNO₃ could successfully inhibits the continuous degradation of electrolytes, preserving uniform and thin CEI layers during the deep cycling. The thickness only slightly increased from 4.8 nm to 5.3 nm, as shown in Figure 4.

We have added these new data as new Fig. 4 on page 16-17 in the revised manuscript.

Fig. 4. Components on the surface of cathodes after cycling. (a) X-ray photoelectron spectroscopy (XPS) profiles of C 1s, F 1s, N 1s and O 1s of CEI formed on NMC811 surface after 5 cycles and (b) XPS depth profiles of O 1s of CEI formed on NMC811 surface after 50 cycles at 0.5 C in Li||NMC811 coin cells with LiFSI/DME and LiFSI-LiNO₃/DME electrolyte. HRTEM analyses of NMC811 cathodes obtained from Li/NMC811 cells using LiFSI/DME (right) and LiFSI-LiNO₃/DME (right) after (c) 5 and (d) 50 cycles.

Comment 7: The authors hypothesize that a thinner CEI created by the LiFSI-LiNO₃-DME electrolyte (Figure 3) contributes to the improved rate performance shown in Figure 2. However,

the performance compared in figure 2 are LiFSI-LiNO₃-DME and LiPF₆-EC-DEC, whereas the systems compared in Figure 3 are LiFSI-LiNO₃-DME and LiFSI-DME. This conflicting experimental design makes this hypothesis impossible to assess. It is therefore recommended once again to compare the NMC 811//Li cell performance between LiFSI-DME and LiFSI-LiNO₃-DME to support this hypothesis.

Response to comment 7: Thank you for your suggestion. We have added extensive experiments to illustrate the key reasons for the improved rate performance. We would like to summarize the key conclusions here.

(1) The XPS analysis of the NMC811 electrode after cycling in LiFSI-LiNO₃/DME and LiFSI/DME is also to verify the above hypothesis that the appearance of LiNO₃ successfully inhibits the continuous degradation of electrolytes, preserving uniform and thin CEI layers during the cycle.

(2) As you suggested, we complement a more adequate and extensive tests between LiFSI-LiNO₃-DME and LiFSI-DME to verify the hypothesis that the LiNO₃-derived EDL enhances the stability of the cathode/electrolyte interface and prevents the continuous decomposition of the ether solvent under high voltage, thereby significantly improving the electrochemical properties including rate performance as can be seen from the Response to your comments 1 and the following comments.

Comment 8: Furthermore, to confirm that the CEI indeed contributes to the rate and low-temperature performance, the following experiment is recommended:

1. Cycle NMC 811 1 time in LiFSI-DME and LiFSI-LiNO₃-DME electrolytes to form the CEI.
2. Disassemble the cells and transfer the cathode into the opposite electrolyte and assess rate/low-temperature performance.

If the CEI is indeed rate limiting, the performance should correspond to the electrolyte the CEI is formed in instead of the final electrolyte applied for cycling

Response to comment 8: We cycled the NMC cathode in the LiFSI-LiNO₃/DME system for 2 times to form CEI, and then replaced the electrolyte with LiFSI/DME, but the reassembled cells cannot operate efficiently, and the capacity decays rapidly after 50 cycles, which strongly demonstrates that the passivation CEI layer alone cannot prevent the decomposition of ether molecules (Supplementary Fig. 19). Our experimental results highlight the critical role of the LiNO₃-containing passivating EDL on improving the cycling stability of high voltage Li/NMC811 cells, which is the central premise of our manuscript.

We have added this new data as Supplementary Fig. 19 in the SI. And the corresponding discussions have also been added on page 15-16 in the revised main text.

Supplementary Fig. 19. Voltage profiles of the Li||NMC811 cell using 1.0 M LiFSI/DME after cycled 10 cycles in LiFSI-LiNO₃/DME. The disassembled NMC811 cathode, Li anode and separator were washed with DME solvent for several times to remove residual lithium salt.

Comment 9: The authors place a significant amount of weight on their MD studies, which supposedly verified their hypothesis. This is not entirely clear to me, however. Specifically, the authors used fixed charges for their atomic model (which was correctly scaled down in order to account for some polarization effects). However, this approach has only been validated in the bulk. How can we be sure that there aren't additional polarization effects near the interface and at the EDL? Application of a polarizable forcefield would go a long way towards settling this, or at the very least the authors should verify that the electrostatic potential of their current setup agrees with QM calculations.

Comment 9: The authors place a significant amount of weight on their MD studies, which supposedly verified their hypothesis. This is not entirely clear to me, however. Specifically, the authors used fixed charges for their atomic model (which was correctly scaled down in order to account for some polarization effects). However, this approach has only been validated in the bulk. How can we be sure that there aren't additional polarization effects near the interface and at the EDL? Application of a polarizable forcefield would go a long way towards settling this, or at the very least the authors should verify that the electrostatic potential of their current setup agrees with QM calculations.

Response to comment 9: Thank you for your valuable suggestions. We agree with the reviewer's viewpoint that although the non-polarized force fields are still adopted for electrolytes by most researchers at present¹⁻⁵, there is no doubt that the polarized force field can more accurately predict thermodynamic, transport and structural properties of electrolytic systems (solvent combined with salts) or ionic complexes (e.g., ionic liquids). However, there are some studies reporting that the turn-off of polarizability of force fields may have minor impact on the

property⁶⁻⁸. To check the impact of polarizability on the non-polarized force fields used in our work, we do follow the reviewer's suggestion to make a comparison between Ab Initio Molecular Dynamics (AIMD) and classical MD simulations of the same system of electrolyte-electrode interfaces.

AIMD based on quantum chemical methods has been shown to be the most accurate analysis of how atoms and molecules move and interact over a fixed time period, and such an accuracy level also requires considerable computational power, which limits the size of the system it can be used to study. At present, a complex and giant system like our electrodes/electrolytes system far exceeds our computational power, thus we compared the properties of the interface obtained by MD and AIMD based on a simplified system (solvent-only system) to verify the reliability of our simulation results. The number density as a function of time and distance from the electrode surface was performed in Supplementary Fig. 26, the evolution of solvent density at the solid/liquid interface obtained by AIMD simulations was found to be very close to that obtained by MD simulations. On further examining the solvent orientation, the angular distributions of the DME adsorbed on the solid/liquid interface, obtained by MD and AIMD, were also very similar (Supplementary Fig. 27). Therefore, we expect that the properties of the entire electrode/electrolyte interface obtained by MD and AIMD simulations will be very similar. Nevertheless, the polarizable forefield is undoubtedly more accurate for molecular dynamic simulation, and we are also trying to develop this method for future work.

Supplementary Fig. 26. Comparison of number densities of pure solvent (DME) obtained from MD and AIMD simulation at 0V.

Supplementary Fig. 27. Comparison of orientation of the DME adsorbed at the electrode obtained from MD and AIMD simulation at 0 V. (a) Schematics for cation orientation. (b) Probability distribution of θ with time.

Reference:

1. Zhou, Y. et al. Real-time mass spectrometric characterization of the solid–electrolyte interphase of a lithium-ion battery. *Nature Nanotechnology* 15, 224-230 (2020).
2. Rakov, D. et al. Stable and Efficient Lithium Metal Anode Cycling through Understanding the Effects of Electrolyte Composition and Electrode Preconditioning. *Chemistry of Materials* 34, 165-177 (2022).
3. Merlet, C. et al. On the molecular origin of supercapacitance in nanoporous carbon electrodes. *Nature Materials* 11, 306-310 (2012).
4. Bi, S. et al. Molecular understanding of charge storage and charging dynamics in supercapacitors with MOF electrodes and ionic liquid electrolytes. *Nature Materials* 19, 552-558 (2020).
5. Rakov, D.A. et al. Engineering high-energy-density sodium battery anodes for improved cycling with superconcentrated ionic-liquid electrolytes. *Nature Materials* 19, 1096-1101 (2020).
6. Bedrov, D., Borodin, O., Li, Z. & Smith, G.D. Influence of Polarization on Structural, Thermodynamic, and Dynamic Properties of Ionic Liquids Obtained from Molecular Dynamics Simulations. *The Journal of Physical Chemistry B* 114, 4984-4997 (2010).
7. Muralidharan, A., Chaudhari, M.I., Pratt, L.R. & Rempe, S.B. Molecular Dynamics of Lithium Ion Transport in a Model Solid Electrolyte Interphase. *Scientific Reports* 8, 10736 (2018).

8. Olivieri, G., Parry, K.M., D'Auria, R., Tobias, D.J. & Brown, M.A. Specific Anion Effects on Na^+ Adsorption at the Aqueous Solution–Air Interface: MD Simulations, SESSA Calculations, and Photoelectron Spectroscopy Experiments. *The Journal of Physical Chemistry B* 122, 910-918 (2018).

Comment 10: Furthermore, the use of fixed charges on the graphene surface to effect “voltage” is suspect [J. Chem. Phys. 141, 184102 (2014)], and can lead to spurious physics. The application of that approach here sows doubt on the central premise: the formation of polymer like layers near the cathode. The authors are encouraged to verify that these motifs survives a more rigorous theoretical examination (namely the constant potential method)

Response to comment 10: Thank you for your valuable suggestions. We agree with the reviewer’s comments. Indeed, constant potential method (CPM) is a more accurate but more computational expansive method to model realistic conditions. However, constant charge method (CCM) is generally considered to be able to simulate open electrode systems at the equilibrium state (where the electrode surface is in contact with the bulk electrolyte¹, for example, a planar², cylindrical³ or spherical⁴ surface), while it may not be suitable to simulate the charging dynamics or nanoporous electrode systems⁵⁻⁷. For example, Holm et al.² proved that in coarse-grained simulation of electrolytes on a smooth electrode surface or atomistically detailed graphene surface, the choice of electrode treatment (constant charge or constant potential approach) did not show a significant effect on electric double layer structure or differential capacitance. Very recently, Feng et al.⁵ reported the first study of constant-potential simulations for exploring the galvanostatic charge–discharge of supercapacitors, and revealed that for the simulation system with electrolytes at the open electrodes (the electrolyte-electrode setup is the same with ours in this work), using CPM and CCM gives nearly the same EDL structure not only in equilibrium situation but also at any time during the charging and discharging process, although showing a significant deviation for nanoporous electrode systems. Specifically, the potential on the flat electrode is computed accordingly under constant charge method, showing a distinctly non-uniform distribution; however, the averaged potential over electrode atoms at each time closely matches the results obtained from the constant potential method. The evolution of cation density at the positive electrode obtained by CCM calculations was also found to be nearly the same as that obtained by CPM, and a comparison of more dynamic EDL structures reaches the same agreement. Therefore, the constant charge method simulation is currently reasonable for the open electrode system that we adopt in this work. Nevertheless, we are also aware of the importance of CPM for modelling realistic conditions, we are trying to perform the CPM simulations for more complex work in the future.

Reference:

1. Vatamanu, J., Borodin, O., Olguin, M., Yushin, G. & Bedrov, D. Charge storage at the nanoscale: understanding the trends from the molecular scale perspective. *J. Mater. Chem. A* 5, 21049–21076 (2017).
2. Breitsprecher, K., Szuttor, K. & Holm, C. Electrode Models for Ionic Liquid-Based Capacitors. *Journal of Physical Chemistry C* 119, 22445-22451 (2015).
3. Yang, L., Fishbine, B. H., Migliori, A. & Pratt, L. R. Molecular simulation of electric double-layer capacitors based on carbon nanotube forests. *J. Am. Chem. Soc.* 131, 12373–12376 (2009).
4. Feng, G., Jiang, D. & Cummings, P. T. Curvature effect on the capacitance of electric double layers at ionic liquid/onion-like carbon interfaces. *J. Chem. Theory Comput.* 8, 1058–1063 (2012).
5. Zeng, L., Wu, T., Ye, T., Mo, T. & Feng, G. Modeling galvanostatic charge-discharge of nanoporous supercapacitors. *Nature Computational Science* 1, 725–731 (2021)
6. Wang, Z., Yang, Y., Olmsted, D.L., Asta, M. & Laird, B.B. Evaluation of the constant potential method in simulating electric double-layer capacitors. *J. Chem. Phys.* 141, 184102 (2014).
7. Péan, C. et al. On the dynamics of charging in nanoporous carbon-based supercapacitors. *ACS Nano* 8, 1576-1583 (2014).

Reviewer #2 (Remarks to the Author):

This paper reports a novel approach that may enlarge the working voltage window toward higher voltage of ether-based electrolytes, such as DME, so that ether-based electrolytes can be used in the LIBs consisting of high-potential layered oxide cathodes and Li metal anode. In this novel approach, designed anions, such as NO_3^- and ClO_4^- , are introduced as additives into ether-based electrolytes to modify the electric-double layer at the oxide cathode surfaces in order to effectively reduce the density of free, unbonded ether solvent molecules, resulting in substantially reduced solvent decomposition at high potential. The paper demonstrates that the successful employment of ether-based electrolytes open up the opportunities for extremely fast charging/discharging and ultra-low temperature applications of high-voltage LIBs. In addition, the paper provides both theoretical calculation and sophisticated surface sensitive analytic data to elucidating the underlying mechanism. The simplicity of the new method, the new application potential, and interesting scientific insight could render this paper having a broad impact in the LIB field.

Response: Thank you very much for your positive comments!

There are a few issues needed to be properly addressed before the paper can be recommended for publication.

Comment 1: *The key point of this paper is the interfacial stability enhancement of the ether-based electrolyte in contact of the high-potential oxide cathode. This paper presents many high-rate data, but they often overlook the interfacial issue due to insufficient reaction time. The authors are suggested to compare the interfacial stability in a quantitative manner among different electrolyte systems. In one method, for example, the authors can run leakage current measurement at selected floating potentials (for example, at 4.3V) for DME electrolytes either with or without NO_3^- and a carbonate electrolyte, which can serve as a benchmark.*

Response to comment 1: We gratefully appreciate for your valuable suggestion. Following your suggestions, we added the following experiments in the revision.

(1) As shown by the new Fig. 2e (shown below), the leakage current as a function of time generated by a potentiostatic hold (4.3V) for cells with different electrolytes. The initial decrease in the curve is the relaxation of the concentration gradient in the cell, and the final static leakage current value is usually used as an indicator to quantify the rate of the parasitic reaction. LiFSI/DME shows a much higher static leakage current than the other two electrolytes, indicating

a faster parasitic reaction between electrode and electrolyte. The presence of NO_3^- significantly improves the stability of the electrode/electrolyte interface, thereby effectively diminishing reaction rate and inhibiting the self-discharge of a fully charged DME-based cells.

(2) The reactivity of the different electrolytes with NMC811 cathodes was further investigated by recording the open circuit voltage (OCV) of a fully charged cells as a function of storage time. As shown in Fig. 2d, the OCV of a fully charged cells with LiFSI/DME drastically decreased after circuit disconnection, whereas a cell charged in LiFSI-LiNO₃/DME exhibited relatively little OCV change, even better than commercial carbonate electrolytes.

We have added these two figures as new Fig. 2d and 2e on page 8 in the revised main text. The corresponding discussion have been added on line 124-138 on page 6-7.

Fig. 2. Comparison of electrochemical properties of LiFSI/DME and LiFSI-LiNO₃/DME. (a) Oxidative stability measured via LSV for Li||Al cells. (b) Rate capability of Li||NMC811 cells under different charging/discharging rates. (c) Long-term cycle performance of Li||NMC811 cells using different electrolytes at 1.0 C rate. (d) Self-discharge tests after a potentiostatic hold at 4.3 V vs Li/Li⁺. (e) Typical current relaxation curves collected from NMC811/Li half-cells during a potentiostatic hold at 4.3 V vs Li/Li⁺.

Comment 2: Also for the same reason, the charge-discharge voltage plots of formation cycle and other cycles of low C-rate (for example, 0.1C) should be provided for comparison.

Response to comment 2: Thank you very much for your valuable suggestions.

(1) The charge-discharge voltage plots of formation cycle were shown in new Supplementary Fig. 2. The coulombic efficiency of LiFSI/DME in formation cycle is only ~ 68.44 %, while that of LiFSI-LiNO₃/DME in the formation cycle is significantly increased to ~ 85.98 %, which further demonstrated that the presence of NO₃⁻ successfully inhibited the side reaction of the DME-based electrolyte at the positive/electrolyte interface. We have added this on page 6 in the revised manuscript.

(2) We have also added the long-cycle performance of NMC811/Li cells using the two electrolytes at a rate of 0.1C (Supplementary Fig. 3), the results show that LiFSI-LiNO₃/DME system can guarantee the interfacial stability even at a low rate, and obtain high Coulombic efficiency (average CE=99.05 % during the first 20 cycles) and stable cycling. On the contrary, serious side reactions occurred at the interface without LiNO₃, resulting in low coulombic efficiency ((average CE= 82.83 % the first 20 cycles) and rapid battery failure.

We have added the figures as new Supplementary Fig. 2-3, and the corresponding discussions were added in lines 109 to 119 on page 6 in the revision.

Supplementary Fig. 2. Formation cycle of Li||NMC811 cells using LiFSI/DME and LiFSI-LiNO₃/DME at 0.2 C rate.

Supplementary Fig. 3. Long-term cycling performance of Li||NMC811 cells using LiFSI/DME and LiFSI-LiNO₃/DME at 0.1 C rate.

Comment 3: In Fig. 2, the paper shows superior cycle stability of NO₃⁻ added DME electrolyte to a carbonate electrolyte. It will be desirable to know if the difference originates solely from the Li metal side and how much the cathode side could contribute, either positively or negatively, to it? The authors may consider to run testing on oxide//oxide symmetric cells to remove the Li metal anode effects.

Response to comment 3: Thank you very much for your valuable suggestions. The first reviewer

suggested using the three-electrode experiment to distinguish the contribution of the positive and negative electrodes to the overpotential. We think the three-electrode experiment can also answer your question. Therefore, the three-electrode experiments are implemented to decouple the anode and cathode contributions to the overpotential at different rates.

As shown in new Supplementary Fig. 11, the overpotential of lithium metal anode in ether-based electrolytes is much smaller than that of carbonate-based electrolytes at low rates (0.2 C, 0.5 C) due to the superior compatibility between lithium metals and ether solvents, but there is no significant difference in the overpotential of the two electrolytes on the cathode side. However, with the increase of the charge/discharge rate, the struggling dynamic behavior of the carbonate-based electrolyte on the cathode side dominates the performance loss. Therefore, the excellent fast charging performance of Li/NMC811 cells with LiFSI-LiNO₃/DME is mainly attributed to the fast kinetic response of our electrolyte on the cathode side.

We have added these new data as new Supplementary Fig. 11 in SI, and the corresponding explanations were added on lines 169-179 on page 9-10 in the revised manuscript.

Supplementary Fig. 11. Voltage curves of the lithium metal anodes (right axis) and NMC811 cathodes (Left axis) with respect to the lithium foil reference electrodes during charge and discharge at (a) 0.2 C (b) 0.5 C (c) 1.0 C (d) 3.0 C (e) 5.0 C current density.

Comment 4: For theoretical calculation, calculation should also be carried out on other “non-improving” anion(s), such as SO_4^{2-} and CO_3^{2-} (Fig. SI 4) for comparison, in order to support, from a “negative” viewpoint, the design principle.

Response to comment 4: Thank you very much for your valuable suggestions. As you suggested, additional calculations were performed using SO_4^{2-} and BF_4^- as additives (It should be pointed out here that due to the problem of the extremely poor solubility of both SO_4^{2-} and CO_3^{2-} in the ether solvent, we decided to replace CO_3^{2-} with BF_4^- to prove our hypothesis). As shown in, the present of SO_4^{2-} and BF_4^- did not significantly increase the proportion of bound-DME at the interface, and the polymer-like chain structure appearing in the NO_3^- system is not observed in the inner-Helmholtz interfacial regions of these two systems, thus further verifying the unique

properties of NO_3^- in changing the EDL structure.

Supplementary Fig. 28. Comparison of number densities of the bound-DME in the interfacial region at different electrolytes system.

Supplementary Fig. 29. Local structure evolution of inner-Helmholtz interfacial regions at cathode surface in 1 M LiFSI-LiNO₃/DME under different voltage.

Supplementary Fig. 30. Snapshots of inner-Helmholtz interfacial regions of the cathode surface in 1 M LiFSI-Li₂SO₄/DME at different voltage.

Comment 5: Other editorial points:

1. Symbols and legends in all figures should be enlarged.
2. The coulombic efficiency scale should be enlarged in all related figures.
3. The color scale in Fig. 4a and b should be explained in caption.
4. Figures 4g and h are very hard to read. Separate blown-up images showing much greater details (such as atom arrangement and bondings) should be given.

Response to comment 5: Thank you so much for your careful check. As you suggested, symbols and legends in all figures have been enlarged. The coulombic efficiency scale has been enlarged in all related figures. The explain of the color scale in Fig. 4a and b have been added in caption. Separate blown-up images with more details are provided as new figures 5e and 5f.

Fig. 5. Molecular dynamics simulation of EDL structure. Number densities of Li^+ in (a) 1 M LiFSI/DME and (b) LiFSI-LiNO₃/DME as a function of distance from the graphite electrode (z) at various potentials (Φ_{EDL}). The color scale indicates Number densities of Li^+ . Comparison of number densities of the (c) Li^+ and (d) bound-DME in the interfacial region at different electrolytes system. Local structure of inner-Helmholtz interfacial regions at cathode surface in (e) 1 M LiFSI/DME and (f) LiFSI-LiNO₃/DME at 0.5 V.

Reviewer #3 (Remarks to the Author):

The authors developed a new electrolyte with stable performance at high voltage and low temperature which is highly desirable for practical applications. They also explained this improved performance based on a stable EDL formed in cathode. The results are very impressive, and the manuscript is well written. Therefore, I recommend publication of this work in NC after addressing the following concerns.

Response: Thank you very much for your positive comments!

Comment 1: One of the most critical properties of Li metal batteries is the coulombic efficiency of electrolyte tested in Li||Cu cells. what is the CE of new electrolytes investigated in this work?

Response to comment 1: Thank you very much for your valuable suggestions. We put the description of lithium metal anode on the supplementary information, please refer to Supplementary Figure 12. It is well known that the ether-based electrolyte is beneficial to enhance cycling of Li metal anode compared to carbonate electrolytes. Here, we found that the LiNO₃ additive could make the electrochemical behavior of Li metal anode even better. Although the coulombic efficiency of the bare DME electrolytes (1 M LiFSI in DME) is as high as 98.5% measured via the method proposed by Adams et al (Supplementary Fig. 12a), the performance of Li||Cu half cell is relatively unstable at long-term cycling (Supplementary Fig. 12b). In contrast, after adding LiNO₃, the Li||Cu half-cell exhibits improved coulombic efficiency (99.1%) and long-term cycling stability with an average CE of 98.86% for 150 cycles.

Supplementary Fig. 12. Li metal performance of the LiFSI-LiNO₃/DME. (a) Aurbach CE [32] of

Li||Cu cells. (b) Cycling CE of Li||Cu cells. (c) Cycling performance of Li||Li cells. Li plating morphology on copper working electrodes in the (d) electrolyte without LiNO₃ and (e) the electrolyte with LiNO₃. The current density is 1 mA cm⁻² and the plating capacity is 1 mAh cm⁻².

Comment 2: *This work used stainless steel coin cell case for electrochemical test. It is known that low concentration LiFSI electrolyte is not stable with stainless steel coin cell case at a voltage higher than 4.2V. are there any corrosion effect observed on the stainless steel coin cell case used in this work? If not, what is the mechanism to avoid the corrosion at low LiFSI concentration?*

Response to comment 2: We gratefully appreciate for your valuable suggestion.

(1) We performed cyclic voltammetry test and SEM observations. It is revealed the introduction of LiNO₃ effectively inhibited the corrosion of stainless steel by low concentration LiFSI electrolytes. For the LiFSI/DME system, a strong side reaction current value was observed in the cyclic voltammetry test when stainless steel was used as the working electrode (new Supplementary Fig. 32). Such a strong Faradic current response is not only derived from the decomposition of the electrolyte but also the contribution of the aluminum corrosion reaction, as shown by the corrosion holes in the SEM picture of the recycled stainless steel (new Supplementary Fig. 33b). In contrast, no apparent response current in the LiFSI / DME system was present within the voltage range of 3-4.5V (new Supplementary Fig. 32), and the cycled stainless steel was as smooth as initially without corrosion pits (new Supplementary Fig. 33c).

For the mechanism, we believe that the constructed Li⁺-rich EDL would not only protect DME from degradation, but also significantly inhibit the attack and corrosion of stainless steel by the free FSI in the solution. On the one hand, there is less free FSI at the Li⁺-rich interface, and most of the DME is bound by Li⁺, which greatly inhibits the dissolution of corrosion products.

We have added this as new Supplementary Fig. 32 and 33, and the corresponding discussions are added on page 32-33 in the Supplementary information.

(2) Inspired by your suggestion and Review 1, to eliminate the effect of corrosive FSI on stainless steel, we further extensively re-compared the electrochemical behavior of LiFSI/DME and LiFSI-LiNO₃/DME by applying an Aluminum coated cathode cap in coin cells. As can be seen from the following added experiments, by applying an Aluminum coated cathode cap, our designed strategy still works and show much better performance. These data excluded the side effect of stainless steel corrosion by FSI and further support our central conclusion.

(a) Both rate and cycling tests ranging from low rate (0.1 C) to high rate (10.0 C) were conducted for NMC811 electrodes in LiFSI/DME and LiFSI-LiNO₃/DME electrolytes. Indeed, 1M LiFSI/DME can be cycled in Li/NMC811 cell for several times after eliminating the factor of stainless steel corrosion. However, As shown in new Fig. 2b in the revised manuscript,

LiFSI/DME suffers rapid capacity fading with increasing current. On the contrary, the LiFSI-LiNO₃/DME exhibits a remarkable rate performance, achieving a discharge capacity of 152 mAh g⁻¹ even at a high rate of 10.0 C. Moreover, the cell can still maintain a stable cycle after returning from the rate of 10.0 C to the rate of 0.5 C, which proves the sustainable adaptability of LiFSI-LiNO₃/DME to the highly active NMC811 cathode material.

(b) The long cycle tests of LiFSI-LiNO₃/DME at different rates are extremely stable (new Fig.2c, new Supplementary Fig. 3-8), while LiFSI/DME is subject to continuous capacity decay at all rates, demonstrating the important role of LiNO₃ in improving oxidation-resistance of DME-based electrolytes.

(c) The reactivity of the different electrolytes with NMC811 cathodes was further investigated by recording the open circuit voltage (OCV) of a fully charged cells as a function of storage time. As shown in new Fig. 2d, the OCV of a fully charged cells with LiFSI/DME drastically decreased after circuit disconnection, whereas a cell charged in LiFSI-LiNO₃/DME exhibited relatively little OCV change, even better than conventional commercial carbonate electrolytes.

(d) The new Fig. 2e shows the leakage current as a function of time generated by a potentiostatic hold (4.3V) for cells with different electrolytes. The initial decrease in the curve is the relaxation of the concentration gradient in the cell, and the final static leakage current value is usually used as an indicator to quantify the rate of the parasitic reaction. LiFSI/DME shows a much higher static leakage current than the other two electrolytes, indicating a faster parasitic reaction between electrode and electrolyte. The presence of NO₃⁻ significantly improves the stability of the electrode/electrolyte interface, thereby effectively diminishing reaction rate and inhibiting the self-discharge of a fully charged DME-based cells.

We have added these new data and corresponding sentences as Fig.2, Supplementary Fig. 2-8 on page 5-8 in the revised main text.

Fig. 2. Comparison of electrochemical properties of LiFSI/DME and LiFSI-LiNO₃/DME. (a) Oxidative stability measured via LSV for Li||Al cells. (b) Rate capability of Li||NMC811 cells under different charging/discharging rates. (c) Long-term cycle performance of Li||NMC811 cells using different electrolytes at 1.0 C rate. (d) Self-discharge tests after a potentiostatic hold at 4.3 V vs Li/Li⁺. (e) Typical current relaxation curves collected from NMC811/Li half-cells during a potentiostatic hold at 4.3 V vs Li/Li⁺.

Supplementary Fig. 3. Long-term cycling performance of Li||NMC811 cells using LiFSI/DME and LiFSI-LiNO₃/DME at 0.1 C rate.

Supplementary Fig. 4. Long-term cycling performance of Li||NMC811 cells using LiFSI/DME and LiFSI-LiNO₃/DME at 1.0 C rate.

Supplementary Fig. 5. Long-term cycling performance of Li||NMC811 cells using LiFSI/DME and LiFSI-LiNO₃/DME at 3.0 C rate.

Supplementary Fig. 6. Long-term cycling performance of Li||NMC811 cells using LiFSI/DME and LiFSI-LiNO₃/DME at 5.0 C rate.

Supplementary Fig. 7. Long-term cycling performance of Li||NMC811 cells using LiFSI/DME and LiFSI-LiNO₃/DME at 10.0 C rate.

Supplementary Fig. 8. Long-term cycling performance of Li||NMC811 cells using LiFSI/DME and LiFSI-LiNO₃/DME at 5.0 C charge and 1.0 C discharge.

Supplementary Fig. 33. Cyclic voltammograms of LiFSI/DME and LiFSI-LiNO₃/DME using stainless steel working electrode in a 2032-coin cell setup. (Working electrode: stainless steel, counter and reference electrode: Li, scan rate: 10 mV/s, Voltage range: 3-4.5V).

Supplementary Fig. 34. SEM images of (a) pristine stainless steel electrode and stainless steel electrode after Cyclic voltammograms test in (b) LiFSI/DME (c) LiFSI-LiNO₃/DME

Comment 3: The author should add the scale bar in the SEM image (Supplementary Fig.5).

Response to comment 3: Thank you so much for your careful check. We have added the scale bar in the SEM image. Please refer to the new Supplementary Fig.10.

Supplementary Fig. 10. Cross-sectional FIB/SEM images of NMC811 particles after 100 cycles in Li||NMC811 cells with (a) 1 M LiFSI/DME and (b) LiFSI-LiNO₃/DME.

Comment 4: In the experimental method, the author mentioned 30 μm of Li metal used for full cell, while in the caption of Fig. 2 and the line of 135 on page 7, the author said the 40 μm of Li used. What is the truth?

Response to comment 4: Thank you so much for your careful check. We are very sorry for the confusion from our mistake. The thickness of the lithium metal we used is 40 μm, and the error has been corrected.

Comment 5: The ionic conductivities for various temperatures of electrolytes samples should be provided.

Response to comment 5: Thank you very much for your valuable suggestions. We have provided the ionic conductivities for various temperatures of electrolytes samples. As shown in Fig.3a, the conductivity of the DME-based electrolyte is observed to retain a remarkable value (3.68 mS cm⁻¹ for LiFSI/DME, 2.87 mS cm⁻¹ for LiFSI-LiNO₃/DME) even at -60 °C are attributed to the low melting point (~ -69 °C), compared with the values less than 0.1 mS cm⁻¹ in LiPF₆/EC+DEC at temperatures below -20 °C. It is noteworthy that only slight variation in the conductivity of the DME-based electrolytes were observed due to the presence of LiNO₃. Moreover, the conductivity of the THF-based electrolyte showed a more significant advantage at ultralow temperature, obtaining a value over 1 ms cm⁻¹ even at -80 °C (2.06 ms cm⁻¹ for LiFSI/THF, 1.22 ms cm⁻¹ for LiFSI- LiNO₃/THF at -80 °C).

We have added these new data as new Fig. 3 and Supplementary Fig. 17, the corresponding

explanations were added on lines 153-159 on page 9 in the revised manuscript.

Fig. 3. Comparison of transport and electrochemical properties between ether-based electrolyte and carbonate-based electrolyte. (a) Conductivity versus temperature. **(b)** Li^+ transference number (τ_{Li^+}) computed from DC polarization measurements at 10 mV using the Bruce–Vincent method. **(c)** Rate capability of Li||NMC811 cells under different charging/discharging rates. **(d)** Long-term super-fast charging/discharging performance of Li||NMC811 cells using different electrolytes at 5.0 C rate. **(e)** the corresponding voltage profiles at different cycles of Li||NMC811 cells using LiFSI-LiNO₃/DME at 5.0 C rate. **(f)** Long-term cycling performances of high-voltage Li||NMC811 full batteries with 40 μm Li anode. The N/P ratios of the Li||NMC811 cell was 2.31. The first two formation cycles were carried out at a 0.1 C rate, followed by 15 cycles at 0.5 C rate, sequential 15 cycles at 1.0 C rate and the long-term cycling was at 2.0 C rate. **(g)** The corresponding voltage profiles of high-voltage Li||NMC811 full batteries using LiFSI-LiNO₃/DME. **(h)** Cycling performance of full cells in LiFSI-LiNO₃/THF

electrolyte at -40 °C and 0.3 C rate. (i) Discharge profiles of Li||NMC811 cells using LiFSI-LiNO₃/THF electrolyte at different temperatures.

Supplementary Fig. 17. Conductivity versus temperature of different electrolytes.

Comment 6: How is the morphologies of Li metal after cycles? The SEM images after cycling will be very helpful for the readers to have a better understanding on the

Response to comment 6: SEM was further employed to observe the morphology of lithium metal deposited in different electrolytes after cycling. Interestingly, LiNO₃ induces Li deposition to grow into a more compact large Li chunks (Supplementary Fig. 12e), which is in great contrast to the porous morphology of Li metal deposited from bare DME electrolyte (Supplementary Fig. 12d). This dendrite-free surface morphology is considered to be a key factor for high Coulombic efficiency. It has been reported that LiNO₃ can participate in the formation of SEI film on the surface of Li metal anode, and the resulting N-rich SEI layer can effectively inhibit the growth of lithium dendrite, thus improving the coulombic efficiency and long cycle life of lithium metal anode.

We have added this on page 12-13 in the Supplementary information.

Supplementary Fig. 12. Li metal performance of the LiFSI-LiNO₃/DME. (a) Aurbach CE ^[32] of Li||Cu cells. (b) Cycling CE of Li||Cu cells. (c) Cycling performance of Li||Li cells. Li plating morphology on copper working electrodes in the d) electrolyte without LiNO₃ and e) the electrolyte with LiNO₃. The current density is 1 mA cm⁻² and the plating capacity is 1 mAh cm⁻².

REVIEWERS' COMMENTS

Reviewer #1 (Remarks to the Author):

This is one of the more in-depth responses that I've seen in terms of the volume of data the authors have been able to produce. I'm satisfied that the authors have addressed my concerns adequately and that the paper is now suitable for publication in Nature Communications

Reviewer #2 (Remarks to the Author):

The authors have conducted new experiments and obtained additional data, which properly addressed all of the previous comments from this reviewer. Overall, this manuscript reports significant new technological advancements for high-rate LIBs along with new scientific insights into the electrochemical aspects of the system. The manuscript is recommended for publication.